# THE RATE-DISTORTION-PERCEPTION TRADE-OFF WITH ALGORITHMIC REALISM

## ABSTRACT

Realism constraints (or constraints on perceptual quality) have received considerable recent attention within the context of lossy compression, particularly of images. Theoretical studies of lossy compression indicate that high-rate common randomness between the compressor and the decompressor is a valuable resource for achieving realism. On the other hand, the utility of significant amounts of common randomness at test time has not been noted in practice. We offer an explanation for this discrepancy by considering a realism constraint that requires satisfying a universal critic that inspects realizations of individual compressed images, or batches thereof. We characterize the optimal rate-distortion-perception trade-off under such a realism constraint, and show that it is asymptotically achievable without any common randomness, unless the batch size is impractically large.

## 1 INTRODUCTION

Realism, or perceptual quality, of reconstructed signals is a long-standing open challenge in lossy compression, particularly for image/video compression (Eckert & Bradley, 1998; Wu et al., 2012). It has received renewed interest in the recent years due to the remarkable progress in image generation models and neural compression techniques. The idea is that reconstructed images should be indistinguishable to humans from naturally occurring ones in addition to having a high pixel-level fidelity to the original source. This ensures that reconstructed images are free of obvious artifacts such as blocking, blurriness, etc.

The idea that the output of the decoder should resemble the source in a statistical sense is not new. Advanced Audio Coding (AAC), for instance, includes a provision to add high-frequency noise to the output so that its power spectrum resembles that of the source (Sayood, 2012). But the idea has received renewed attention with the emergence of adversarial loss functions in learned compression (Santurkar et al., 2018; Tschannen et al., 2018; Agustsson et al., 2019; Blau & Michaeli, 2019). In practice, this has proven to be a powerful method for ensuring that reconstructed images have high perceptual quality (Agustsson et al., 2019; Mentzer et al., 2020; He et al., 2022a; Iwai et al., 2024). Adversarial loss functions can in many cases be viewed as variational forms of statistical divergences. Thus one can think of constraining the distribution of reconstructions to be close to that of the source according to some divergence, in addition to requiring that each reconstructed image be close to its respective source according to conventional notions of distortion.

Rate-distortion theory characterizes the optimal trade-off between rate and distortion in lossy compression (Pearlman & Said, 2011; Sayood, 2012). The fundamental object in the theory is the *rate-distortion function*, for a given source distribution $p_X$ :

$$\Delta \in [0, \infty) \mapsto R^{(0)}(\Delta) := \min_{\substack{p_{Y|X} \text{ s.t.} \\ \mathbb{E}_p[d(X,Y)] \leq \Delta}} I_p(X;Y), \tag{1}$$

where $p_{X,Y}$ is defined as $p_X \cdot p_{Y|X}$. This function has been shown to describe the optimal trade-off between rate and distortion under a variety of assumptions. Blau & Michaeli (2019) postulated an augmented form that includes a *distribution matching* constraint, which they call the *rate-distortion-perception* (RDP) function

$$(\Delta, \lambda) \in [0, \infty)^2 \mapsto R^{(1)}(\Delta, \lambda) := \min_{\substack{p_{Y|X} \text{ s.t.} \\ \mathcal{D}(p_X, p_Y) \leq \lambda, \\ \mathbb{E}_p[d(X,Y)] \leq \Delta}} I_p(X;Y), \tag{2}$$

where $\mathcal{D}$ can be any divergence between distributions. This function has likewise been shown to describe the optimal trade-off between rate, distortion, and realism under a variety of assumptions (Theis & Wagner; Chen et al., 2022). Curiously, however, these results show that substantial amounts of high-quality common randomness are needed to meet the $R^{(1)}(\cdot, 0)$ bound (Saldi et al., 2015; Wagner, 2022; Chen et al., 2022) (see also Xu et al. (2023)). The exception is the case in which the realism constraint is imposed in a very weak form, namely that the histograms of the source and reconstruction images should be close on a per-realization basis (Chen et al., 2022). Note that common pseudorandomness, say generated from a shared seed, does not qualify as common randomness for the purposes of the above results.

On the other hand, the theoretical prediction that lossy compression schemes would benefit from substantial amounts of high-quality common randomness between the encoder and decoder has not been observed in practice. To the best of our knowledge, there exist compression schemes (Agustsson et al., 2023; He et al., 2022a; Hoogeboom et al., 2023; Ghouse et al., 2023; Mentzer et al., 2020; Yang & Mandt, 2023), considered as state-of-the-art, that do not involve any common randomness. While it is possible that future designs will find common randomness to be a valuable resource, it seems more likely that the discrepancy between the theoretical prediction and practical experience lies with a flaw with the theoretical models.

Consider a communication system for which a strong realism constraint is imposed: the distribution of the reconstructions must be close to the distribution of natural images, say, in Wasserstein or total variation distance (TVD). If the source distribution is continuous, then the code cannot be deterministic, for otherwise the reconstruction distribution would be supported on a countable set (corresponding to the set of received bit strings). Thus some amount of randomization is required to meet the constraint. The decoder can randomize its output in a way that "spreads" the point masses out to form a continuous distribution, but adding independent noise at the decoder inevitably degrades the distortion. Common randomness is useful because it allows the discrete reconstruction points to be dispersed to form a continuous distribution without less overall distortion. This is the basis for the finding that common randomness is a useful resource for compression under realism constraints (Theis & Agustsson, 2021).[1]

The above reasoning is evidently sensitive to the nature of the realism constraint. If we simply require that each reconstructed image appear realistic in its own right, without reference to the reconstruction ensemble, then the spreading process mentioned above is unnecessary. It follows that there would be no need for randomization. This is relevant because human observers, who are the ultimate arbiters of realism in practice, are adept at identifying unrealistic features of individual images. Yet it is difficult for human observers to distinguish between a continuous ensemble and one that is discrete with a very large support set, since doing so would require viewing (and remembering) many images. In short, human critics are very good at spotting unrealistic aspects of individual images but are expected to be poor at detecting subtle ensemble-level differences.

This suggests posing the realism constraint in a way that better captures the relative strengths and weaknesses of human critics. The aforementioned strong realism constraint has also been challenged in the context of other problems, such as generative modeling (Theis, 2024). We consider a novel formulation of the lossy compression problem in which the goal is to satisfy a critic that is incredibly discriminating when viewing individual images. In fact, a reconstructed image is declared unrealistic if there exists some computable test, no matter how complex, that can distinguish it from the set of typical source images (see Definition 3.5 to follow). At the same time, we assume that the critic can glean information about the ensemble only by inspecting batches of individual samples. Under this formulation, we show that the rate-distortion-perception function $R^{(1)}(\cdot, 0)$ in (2) is achievable without common randomness unless the batch size is unreasonably high—on par with the number of possible outputs of the decoder (Theorems 4.1 and 4.2). If common randomness is not needed to fool this critic, it should not be needed to fool any weaker (and more practical) critic, since the stronger critic subsumes the weaker one. This is akin to how in cryptography one might prove security guarantees assuming a very strong adversary, stronger than can be implemented in practice. The fact that the adversary cannot be practically implemented is a strength of our approach. It is notable that there exist compressors that can satisfy such discriminating critics at all. It is all the more notable that

---

[1]It is now apparent why sharing a pseudorandom seed is insufficient, as this would expand the number of distinct reproductions by a multiplicative factor equal to the number of possible values of the seed, which is relatively small if the seed is short.

such critics can be satisfied while achieving the rate-distortion-perception function $R^{(1)}(\cdot, 0)$ in (2), which is the most optimistic rate-distortion trade-off possible under the circumstances. Conversely, we show that common randomness is indeed beneficial if the batch size is extremely large, larger than would ever occur in practice (Theorem 4.4). In this regime, our realism measure reduces to a divergence and common randomness is again useful. These two results clarify that common randomness is indeed useful, consistent with theoretical predictions, but only in regimes that do not occur in practice, consistent with the current state of the experimental literature. Our results show the existence of optimal schemes which do not involve any common randomness at test time, but there may exist other optimal schemes, which rely on common randomness at test time, as well as learned schemes relying on common randomness at training time.

In Section 2, we provide some background on the formalism for critics in algorithmic information theory. In Section 3, we introduce our new formalism for the RDP trade-off. In Section 4, we state our main results, namely Theorems 4.1, 4.2, and 4.4. All proofs are deferred to the appendices.

## 2 BACKGROUND

### 2.1 NOTATION

Calligraphic letters such as $\mathcal{X}$ denote sets, except in $p_{\mathcal{J}}^{\mathcal{U}}$, which denotes the uniform distribution over set $\mathcal{J}$. The cardinality of a finite set $\mathcal{X}$ is denoted $|\mathcal{X}|$. We denote by $[a]$ the set $\{1, ..., \lfloor a \rfloor\}$ and by $\{0, 1\}^*$ the set of non-empty finite strings of 0's and 1's. Given a real number $\tau$, we denote by $\lfloor \tau \rceil$ (resp. $\lceil \tau \rceil$) the largest (resp. smallest) integer less (resp. greater) than or equal to $\tau$. We use $x_{1:n}$ to denote a finite sequence $(x_1, ..., x_n)$, and $\mathbf{x}^{(n,b)}$ to denote a batch $\{x_{1:n}^{(k)}\}_{k \in [b]}$ of $b$ strings, each being of length $n$. We abbreviate $\mathbf{x}^{(1,b)}$ with $\mathbf{x}^{(b)}$. The length of a string $x$ is denoted by $l(x)$.

We denote the set of (strictly) positive reals by $\mathbb{R}_+$, the set of (strictly) positive integers by $\mathbb{N}$, the set of rational numbers by $\mathbb{Q}$, and the Borel $\sigma$-algebra of $\mathbb{R}$ by $\mathcal{B}(\mathbb{R})$. The closure of a set $\mathcal{A}$ is denoted by $cl(\mathcal{A})$. We use $\equiv$ to denote equality of distributions, and $I_p(X; Y)$ to denote the mutual information between random variables $X$ and $Y$ with respect to joint distribution $p_{X,Y}$. Logarithms are in base 2. The total variation distance between distributions $p$ and $q$ on a finite set $\mathcal{X}$ is defined by

$$\|p - q\|_{TV} := \frac{1}{2} \sum_{x \in \mathcal{X}} |p(x) - q(x)|.$$

For any nonempty finite set $\mathcal{X}$, and any distribution $p$ on $\mathcal{X}$, we denote by $p^{\otimes *}$ the function defined on $\{0, 1\}^*$, which is null outside of $\cup_{n \in \mathbb{N}} \mathcal{X}^n$, and such that for every $n \in \mathbb{N}$, the restriction of $p^{\otimes *}$ on $\mathcal{X}^n$ is $p^{\otimes n}$. For a finite set $\mathcal{X}$, the empirical distribution of a sequence $x_{1:n} \in \mathcal{X}^n$ is denoted $\mathbb{P}_{\mathcal{X}}^{\text{emp}}(x_{1:n})$. Given a distribution $P_{X_{1:n}}$ on $\mathcal{X}^n$, we denote by $\hat{P}_{\mathcal{X}}[X_{1:n}]$ the *average marginal distribution* of random string $X_{1:n}$, i.e., the distribution on $\mathcal{X}$ defined by:

$$\hat{P}_{\mathcal{X}}[X_{1:n}] := \frac{1}{n} \sum_{t=1}^{n} P_{X_t}.$$

### 2.2 LOSSY COMPRESSION ALGORITHMS WITHOUT COMMON RANDOMNESS

The performance of practical lossy compression schemes in terms of realism (or perceptual quality) is generally measured with well established metrics such as FID (Heusel et al., 2017), LPIPS (Zhang et al., 2018), PieAPP (Prashnani et al., 2018), and DISTS (Ding et al., 2022). Distortion is often measured with PSNR. According to these metrics, the following lossy compression algorithms are state-of-the-art. In particular, these schemes achieve visually pleasing reconstructions at very low compression rates. None of these algorithms make use of common randomness. The schemes in Mentzer et al. (2020), He et al. (2022a), and Agustsson et al. (2023) were obtained by training with an adversarial loss, a method inspired from generative adversarial networks (GANs). The former combines a conditional GAN with the scale hyperprior method of Ballé et al. (2018). The latter is an extension of the ELIC scheme (He et al., 2022b), which is state-of-the-art in terms of rate and distortion. The loss function of the latter was augmented, in particular, with an adversarial term and an LPIPS term. The method proposed in Agustsson et al. (2023) is inspired from He et al. (2022b) and Mentzer et al. (2020). The schemes in Yang & Mandt (2023), Ghouse et al. (2023), and

Hoogeboom et al. (2023) rely on diffusion models. The first uses a diffusion model conditioned on quantized latents. The two other schemes first train an autoencoder for rate and distortion, then train a diffusion model which improves the visual quality of the latter's output. The fact that none of these state-of-the-art algorithms make use of common randomness supports the theoretical results derived in the present paper.

### 2.3 BACKGROUND ON ALGORITHMIC INFORMATION THEORY

The theory of $p$-critics and universal critics has recently been brought to the attention of the machine vision community via Theis (2024). We refer to it for readers interested in a high-level and insightful presentation of the topic and its usefulness in diverse machine learning tasks (generative modeling, outlier detection). Relevant background on computability theory is provided in Appendix A. Throughout the paper, we assume that the source $X$ follows a distribution $p_X$ on a finite set $\mathcal{X}$, and that $p_X$ is a computable function from $\mathcal{X}$ to $(0, 1)$. We identify every element of $\mathcal{X}$ with a string of 0's and 1's, via an injection from $\mathcal{X}$ to $\{0, 1\}^s$, for some $s \in \mathbb{N}$. For example, if $\mathcal{X}$ is a set of images of a given resolution, then one can identify each image with the corresponding output from a fixed-length lossless compressor. The following definition is substantially close to Li & Vitányi (2019, Definition 4.3.8). See also in Li & Vitányi (2019, Lemma 4.3.5).

**Definition 2.1.** *Consider a finite set $\mathcal{X}$, identified with a subset of $\{0, 1\}^s$. Let $p$ be a distribution on $\mathcal{X}$ such that $\forall x \in \mathcal{X}, p(x) > 0$. A $p$-critic is a function $\delta : \mathcal{X} \to \mathbb{R}$, such that*

$$\sum_{x \in \mathcal{X}} p(x) 2^{\delta(x)} \leq 1. \tag{3}$$

*A $p^{\otimes *}$-critic is a function $\delta : \cup_{n \in \mathbb{N}} \mathcal{X}^n \to \mathbb{R}$, such that for every input dimension $n \in \mathbb{N}$, we have*

$$\sum_{x \in \mathcal{X}^n} p^{\otimes n}(x) 2^{\delta(x)} \leq 1. \tag{4}$$

The notion of $p^{\otimes *}$-critic in Definition 2.1 is used to study an asymptotic regime in Section 3.2. Note that for any probability distribution $\pi$ on $\mathbb{N}$, the mixture $\tilde{p} := \sum_{n \in \mathbb{N}} \pi(n) p^{\otimes n}$ is a probability measure. By multiplying (4) by $\pi_n$, and summing over $n$, we obtain

$$\sum_{x \in \cup_{n \in \mathbb{N}} \mathcal{X}^n} \tilde{p}(x) 2^{\delta(x)} \leq 1. \tag{5}$$

Hence, a $p$-critic (resp. $p^{\otimes *}$-critic) is akin to a log-likelihood ratio: given a $p$-critic (resp. $p^{\otimes *}$-critic) $\delta$, setting $q : x \mapsto p(x) 2^{\delta(x)}$ (resp. $q : x \mapsto \tilde{p}(x) 2^{\delta(x)}$) gives

$$\forall x \in \mathcal{X} \text{ s.t. } p(x) > 0, \ \delta(x) = \log\left(\frac{q(x)}{p(x)}\right) \text{ (resp. } \log\left(\frac{q(x)}{\tilde{p}(x)}\right)\text{)}, \quad \text{and} \quad \sum_{x \in \mathcal{X}} q(x) \leq 1. \tag{6}$$

Links to hypothesis testing are discussed in Theis (2024), where a sample $x$ is deemed unrealistic if the likelihood ratio is large enough. Hence, intuitively, $\delta(x)$ can be considered as a measure of *realism deficiency* of $x$. The strength of this theory lies in the existence of objects (critics, measures) having a so-called *universality property*. For the purpose of clarity, we defer such results to Appendix A, as they are only used in our proofs.

## 3 NEW MODEL FOR THE RATE-DISTORTION-PERCEPTION TRADE-OFF

### 3.1 THE ONE-SHOT SETTING

We consider a function $d : \mathcal{X} \times \mathcal{X} \to [0, \infty)$ called the distortion measure. A compression scheme can be randomized, and potentially leverage available common randomness $J$ between the encoder and the decoder, as depicted in Figure 1 and formalized in the following definition.

**Definition 3.1.** *Given non-negative reals $R$ and $R_c$, an $(R, R_c)$ code is a privately randomized encoder and decoder couple $(F, G)$ consisting of a conditional distribution $F_{M|X,J}$ from $\mathcal{X} \times [2^{R_c}]$ to $[2^R]$, and a conditional distribution $G_{Y|M,J}$ from $[2^R] \times [2^{R_c}]$ to $\mathcal{X}$. Variables $M$ and $Y$ are called the message and reconstruction, respectively, and distribution*

$$P := p_X \cdot p_{[2^{R_c}]}^{\mathcal{U}} \cdot F_{M|X,J} \cdot G_{Y|M,J}$$

Figure 1: The system model for the one-shot setting.

*is called the distribution induced by the code. Moreover, such a code is said to be deterministic if $R_c = 0$ and mappings $F, G$ are deterministic.*

We propose a new RDP trade-off, formalized in the following two definitions.

**Definition 3.2.** *We extend $d$ into an additive distortion measure on batches of elements of $\mathcal{X}$: for all $B \in \mathbb{N}$,*

$$\forall (\mathbf{x}^{(B)}, \mathbf{y}^{(B)}) \in \mathcal{X}^B \times \mathcal{X}^B, \quad d(\mathbf{x}^{(B)}, \mathbf{y}^{(B)}) := \tfrac{1}{B}\sum_{k=1}^{B} d(x^{(k)}, y^{(k)}).$$

**Definition 3.3.** *Consider a positive integer $B$, and a $p_X^{\otimes B}$-critic $\delta$. A tuple $(R, \Delta, C)$ is said to be $\delta$-achievable with algorithmic realism if there exists some $R_c \in \mathbb{R}_{\geq 0}$ and an $(R, R_c)$ code such that the distribution $P$ induced by the code satisfies*

$$\mathbb{E}_{P^{\otimes B}}\big[d(\mathbf{X}^{(B)}, \mathbf{Y}^{(B)})\big] \leq \Delta \ \text{ and} \tag{7}$$

$$\mathbb{E}_{P^{\otimes B}}[\delta(\mathbf{Y}^{(B)})] \leq C, \tag{8}$$

*where $\mathbf{X}^{(B)}$ denotes a batch of $B$ i.i.d. source samples, and $\mathbf{Y}^{(B)}$ the batch of corresponding reconstructions produced by the code (with each source sample being compressed separately). If the code is deterministic, then we say that $(R, \Delta, C)$ is $\delta$-achievable with a deterministic code.*

The main difference with the original RDP trade-off of Blau & Michaeli (2019) pertains to the realism constraint. In the latter formulation, the realism constraint is $\mathcal{D}(p_X, P_Y) \leq C$, where $\mathcal{D}$ is some divergence. Intuitively, that constraint corresponds to the special case of infinite batch size in the RDP trade-off proposed in Definition 3.3, since the discrete distributions $p_X$ and $P_Y$ can be approximated arbitrarily well using a large enough number of samples. In that sense, our proposed RDP framework generalizes the original one, through involving elements of practical realism metrics, such as the number $B$ of samples which are inspected, and a scoring function $\delta$ which is required to be approximable via an algorithm. Theorem 4.4 to follow constitutes a rigorous statement of this intuition. We provide achievable points in the sense of Definition 3.3 in Section 4.2. In the next section, we define an asymptotic notion of achievability.

### 3.2 ASYMPTOTIC SETTING

In order to derive insight into the corresponding RDP trade-off, we study a special case, which is typical in the information theory literature. We consider the compression of a source distributed according to $p_X^{\otimes n}$, with $n$ a large integer. More precisely, we study the RDP trade-off in asymptotic settings where both $n$ and the batch size go to infinity.

The extension of $d$ into an *additive distortion measure* on finite sequences, and batches of finite sequences, follows from Definition 3.2. The setup is depicted in Figure 2. Given a coding scheme, each item in a batch of source samples is compressed separately, and realism is measured based on the resulting batch of reconstructions. This is formalized in the definition below.

**Definition 3.4.** *Given $R, R_c \geq 0$, and $n \in \mathbb{N}$, a $(n, R, R_c)$ code is a privately randomized encoder and decoder couple $(F^{(n)}, G^{(n)})$ consisting of a mapping $F^{(n)}_{M|X_{1:n}, J}$ from $\mathcal{X}^n \times [2^{nR_c}]$ to $[2^{nR}]$ and a mapping $G^{(n)}_{Y_{1:n}|M, J}$ from $[2^{nR_c}] \times [2^{nR}]$ to $\mathcal{X}^n$. Moreover, such a code is said to be fully deterministic if $R_c = 0$ and both $F^{(n)}$ and $G^{(n)}$ are deterministic. The distribution induced by the code is*

$$P^{(n)} := p_X^{\otimes n} \cdot p_{[2^{nR_c}]}^{\mathcal{U}} \cdot F^{(n)}_{M|X_{1:n}, J} \cdot G^{(n)}_{Y_{1:n}|M, J},$$

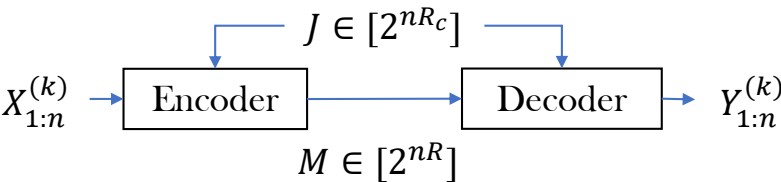

Figure 2: The system model for the asymptotic setting. Index $k$ ranges from 1 to the batch size. The same encoder-decoder pair is used to process each source sample in the batch.

*and variable $Y_{1:n}$ is called the reconstruction.*

We define asymptotic achievability as follows. See Appendix A for background on notions of computability.

**Definition 3.5.**
*A quadruplet $(R, R_c, \{B_n\}_{n \geq 1}, \Delta)$ is said to be asymptotically achievable with algorithmic realism if for any $\varepsilon > 0$, there exists a sequence of codes $\{(F^{(n)}, G^{(n)})\}_n$, the $n$-th being $(n, R + \varepsilon, R_c)$, such that the sequence $\{P^{(n)}\}_n$ of distributions induced by the codes satisfies*

$$\limsup_{n \to \infty} \mathbb{E}_{(P^{(n)})^{\otimes B_n}} \left[ d(\mathbf{X}^{(n, B_n)}, \mathbf{Y}^{(n, B_n)}) \right] \leq \Delta + \varepsilon, \tag{9}$$

*and for any lower semi-computable $p_X^{\otimes *}$-critic $\delta$,*

$$\sup_{n \in \mathbb{N}} \mathbb{E}_{(P^{(n)})^{\otimes B_n}} \left[ \delta(\mathbf{Y}^{(n, B_n)}) \right] < \infty. \tag{10}$$

*We say that $(R, \{B_n\}_{n \geq 1}, \Delta)$ is achievable with a fully deterministic scheme if for each $n$, the code $(F^{(n)}, G^{(n)})$ is fully deterministic.*

Constraint (10) is very stringent: a single compression scheme is to satisfy a performance guarantee for every lower semi-computable $p_X^{\otimes *}$-critic (i.e. every relevant one). The motivation for the specific form of (10) is firstly from the algorithmic information theory literature: it is stated in Li & Vitányi (2019, p.140) that a sample from a large set, identified to a long string of 0's and 1's of some length $k$, is realistic if its realism deficiency is small compared to $k$. The constraint in (10) is at least as stringent, since in our asymptotic setting, each $x_{1:n} \in \mathcal{X}^n$ is identified with a string of length linear in $n$, while we require the realism deficiency to be bounded. Moreover, consider the following simple example. Assume $\mathcal{X} = \{0, 1\}$, and $p_X$ is a Bernoulli distribution $B(\rho)$. Consider the 0-1 distortion (also called Hamming distortion), and some distortion level $\Delta < \min(\rho, 1 - \rho)$. Then, for large enough $n$, the classical rate-distortion optimal code appearing in the information theory literature produces reconstructions having a frequency of 1's of roughly $(\rho - \Delta)/(1 - 2\Delta)$ (Cover & Thomas, 2006, Sections 10.3.1 and 10.5), i.e. different from $\rho$ (if $\rho \neq 1/2$ and $\Delta > 0$). Then, for the $p_X^{\otimes *}$-critic appearing in Appendix G (Claim G.1), which involves the frequency of occurrence of a pattern, the expected score diverges as $n$ goes to infinity. Hence, the constraint in (10) is not satisfied by such a code, optimized only for rate and distortion, but not for realism. This concludes the definitions for our setup. In the next sections, we present our results, in the one-shot setting and in asymptotic settings.

## 4 RESULTS

### 4.1 LOW BATCH SIZE REGIME

The following theorem states that $R^{(1)}(\cdot, 0)$, defined in (2), which naturaly arises in the distribution matching formalism, also characterizes the optimal trade-off in our asymptotic setting, when the batch size is not impractically large.

**Theorem 4.1.** *Consider a sequence $\{B_n\}_{n \geq 1}$ of positive integers such that*

$$\log(B_n)/n \xrightarrow[n \to \infty]{} 0. \tag{11}$$

*For any $\Delta \in \mathbb{R}_+$, let $R(\Delta)$ be the infimum of rates $R$ such that there exists $R_c \in \mathbb{R}_{\geq 0}$ such that $(R, R_c, \{B_n\}_{n \geq 1}, \Delta)$ is asymptotically achievable with algorithmic realism. Moreover, for*

*any $\Delta \in \mathbb{R}_+$, let $R_*(\Delta)$ be the infimum of rates $R$ such that $(R, \{B_n\}_{n \geq 1}, \Delta)$ is asymptotically achievable with algorithmic realism with fully deterministic codes. Then, we have*

$$\forall \Delta \in \mathbb{R}_+ \text{ s.t. } R^{(1)}(\Delta, 0) < H_p(X), \text{ we have } R(\Delta) = R_*(\Delta) = R^{(1)}(\Delta, 0). \tag{12}$$

The proof is provided in Appendices C and D. The strength of this result lies in how stringent constraint (10) is: a single compression scheme satisfies a performance guarantee for every relevant $p_X^{\otimes *}$-critic, and deterministic schemes are sufficient. Moreover, one can find such a scheme for any batch size sequence which is sub-exponential in the dimension $n$ of the source, i.e. for all regimes where the batch size is not impractically large. To prove the achievability direction of Theorem 4.1, we leverage the existence of a *universal $p_X^{\otimes *}$-critic $\delta_0$* (see Appendix A.2), which is one of the great successes of algorithmic information theory. Indeed, it is sufficient to construct a scheme which achieves (10) only for such a $\delta_0$, which is more sensitive than all relevant $p_X^{\otimes *}$-critics. It is a very strong critic, stronger than can be implemented in practice, which is another strength of Theorem 4.1.

## 4.2 ONE-SHOT ACHIEVABLE POINTS

For theoretical interest, we provide a family of points which are achievable, in the sense of Definition 3.3, without any statistical assumption on the source distribution $p_X$. For the sake of gleaning intuition, one can consider the following example.

- $\mathcal{X}$ is a finite set of images, e.g. the set of all images of a given resolution, with a finite range for pixels (finite precision).
- $d$ is the mean squared error between pixel values.
- $B$ is the number of images inspected by the critic at a time.
- $R_1$ is the number of bits into which a given image is compressed.

**Theorem 4.2.** *Consider a finite set $\mathcal{X}$ such that $|\mathcal{X}| \geq 2$, a computable distribution $p_X$ on $\mathcal{X}$ such that $\forall x \in \mathcal{X}, p_X(x) > 0$, a positive integer $B$, some $R > \log(B)/\log(\mathcal{X})$, some $\Delta \in \mathbb{R}_+$, and a $p_X^{\otimes B}$-critic $\delta$. Consider any conditional transition kernel $p_{Y|X}$ from $\mathcal{X}$ to $\mathcal{X}$ satisfying*

$$p_Y \equiv p_X, \quad \mathbb{E}_p[d(X, Y)] \leq \Delta. \tag{13}$$

*Then, for any $\varepsilon \in (0, \Delta/2)$, and any $\gamma > 0$, the triplet $(R_1, \Delta_1, C_1)$ is $\delta$-achievable, with a $(R_1, 0)$ code, where*

$$R_1 := R \log(|\mathcal{X}|) \tag{14}$$

$$\Delta_1 := \Delta + \varepsilon + \frac{6\Delta}{\varepsilon} \max(d) \cdot \eta_{R,\gamma} \tag{15}$$

$$C_1 := \frac{3\Delta}{\varepsilon} \left[ \frac{B^2}{\lfloor 2^{R_1} \rfloor} + 2B\eta_{R,\gamma} \right] \cdot \max_x B \log \frac{1}{p_X(x)} \tag{16}$$

$$\eta_{R,\gamma} := p(\mathcal{A}_{R,\gamma}) + 2^{-\gamma \log(|\mathcal{X}|)/2} \tag{17}$$

$$\mathcal{A}_{R,\gamma} := \left\{ (x, y) \in \mathcal{X}^2 \mid \log \left( \frac{p_{X,Y}(x,y)}{p_X(x) p_Y(y)} \right) - \log(\lfloor 2^{R_1} \rfloor) > -\gamma \log(|\mathcal{X}|) \right\}, \tag{18}$$

*with the convention $0/0 := 1$.*

The proof is provided in Appendix B. The term $B^2/\lfloor 2^{R_1} \rfloor$ is an upper bound on the probability that two source samples in the batch are compressed into the same message. This is related to the so-called *birthday paradox* (see Appendix I). The term $\max_x B \log(1/p_X(x))$ is an upper bound on the output of $\delta$, which follows from Definition 2.1.

Theorem 4.2 provides insights on the asymptotic regime of Theorem 4.1. Consider the limit of large $|\mathcal{X}|$, with fixed $R, \Delta, \varepsilon, \gamma$, and with $\log(B) = o(\log|\mathcal{X}|)$. We know that

$$\mathbb{E}_p \left[ \log \left( \frac{p_{X,Y}(x,y)}{p_X(x) p_Y(y)} \right) \right] = I_p(X; Y). \tag{19}$$

Hence, if this log-likelihood ratio concentrates well, and if $R_1 > I_p(X; Y)$, as in the definition of $R^{(1)}(\cdot, 0)$ in (2), then $p(\mathcal{A}_{R,\gamma})$ is small for small enough $\gamma$. In such an asymptotic regime, we obtain

$\Delta_1 \approx \Delta$, and $C_1 = O(1)$. Therefore, the assumption in Theorem 4.1, that the source is of the form $p_X^{\otimes n}$ for some large $n$, is only used to ensure fast concentration of the log-likelihood ratio. Hence, Theorem 4.1 can be extended to a larger set of sources. In the next section, we present our last main result, which pertains to an asymptotic regime with large batch size.

### 4.3 GENERALIZING THE DISTRIBUTION MATCHING FORMALISM

In this section, we present a result which connects our proposed formalism for the RDP trade-off to the distribution matching formalism of Blau & Michaeli (2019), and concludes our findings regarding the role of common randomness.

#### 4.3.1 BACKGROUND

Under the distribution matching formalism for the RDP trade-off, the natural asymptotic notion of achievability is as follows.

**Definition 4.3.** *(Saldi et al., 2015; Blau & Michaeli, 2019)*
*A quadruplet $(R, R_c, \{B_n\}_{n \geq 1}, \Delta)$ is said to be asymptotically achievable with near-perfect realism if for any $\varepsilon > 0$, there exists a sequence of codes $\{(F^{(n)}, G^{(n)})\}_n$, the $n$-th being $(n, R + \varepsilon, R_c)$, such that the sequence $\{P^{(n)}\}_n$ of distributions induced by the codes satisfies*

$$\limsup_{n \to \infty} \mathbb{E}_{P^{(n)}}\left[d(X_{1:n}, Y_{1:n})\right] \leq \Delta + \varepsilon,$$

$$\|P_{Y_{1:n}}^{(n)} - p_X^{\otimes n}\|_{TV} \xrightarrow[n \to \infty]{} 0. \tag{20}$$

The TVD in (20) is directly related to the performance of the optimal hypothesis tester between the reconstruction distribution $P_{Y_{1:n}}^{(n)}$, and the source distribution $p_X^{\otimes n}$ (Blau & Michaeli, 2019).

Replacing (20) with

$$\exists N \in \mathbb{N}, \forall n \geq N, \ P_{Y_{1:n}}^{(n)} \equiv p_X^{\otimes n} \tag{21}$$

gives the notion of asymptotic *achievability with perfect realism*. It was shown that these two notions are equivalent for finite-valued sources (Saldi et al., 2015), as well as for continuous sources under mild assumptions (Saldi et al., 2015; Wagner, 2022).

#### 4.3.2 CONNECTION TO OUR FORMALISM

As stated in the theorem below, in a certain large batch size regime, asymptotic achievability with algorithmic realism (Definition 3.5) is equivalent to asymptotic achievability with near-perfect realism (Definition 4.3). The proof is provided in Appendix E.

**Theorem 4.4.** *Consider a computable increasing sequence $\{B_n\}_{n \geq 1}$ of positive integers such that*

$$\frac{B_n}{|\mathcal{X}|^n} \to \infty. \tag{22}$$

*Then, for any $R_c \in \mathbb{R}_{\geq 0}$, and any $(R, \Delta) \in (\mathbb{R}_+)^2$, tuple $(R, R_c, \{B_n\}_{n \geq 1}, \Delta)$ is asymptotically achievable with algorithmic realism if and only if $(R, R_c, \Delta)$ is asymptotically achievable with near-perfect realism, if and only if $(R, R_c, \Delta)$ is asymptotically achievable with perfect realism.*

Hence, Theorem 4.4, similarly to the finding in Theis (2024), shows that for large batch size, our formalism is equivalent to the distribution matching formalism. Hence, the former is a generalization of the latter. Moreover, Theorem 4.4 and prior work on the distribution matching formalism (Saldi et al., 2015; Wagner, 2022; Chen et al., 2022) imply that common randomness is useful when the size of the batch inspected by the critic is extremely large.

## 5 DISCUSSION

Theorem 4.1 states that common randomness does not improve the trade-off under our formalism, in all regimes where the batch size is not impractically large with respect to the dimension $n$ of the

source. Theorem 4.4 states that common randomness is useful — consistent with prior theoretical predictions — when the batch size is extremely large. Thus, Theorems 4.1 and 4.4 indicate that, in order to understand the role of randomization in lossy compression with realism constraints, the focus should be shifted to the size of the batch inspected by the critic. A continuation of our work could be to investigate realism metrics, where particular attention would be given to the choice of the batch size. This could lead to highlighting specific strengths and weaknesses of existing realism metrics. It may also inspire a critical assessment of the relative performance of existing compression schemes, depending on the choice of realism metric. Another continuation could be to more precisely characterize the amount of randomness needed as a function of the batch size. Furthermore, possible extensions of our setup include compression with side information, and other distributed settings.

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

# A FURTHER BACKGROUND ON ALGORITHMIC INFORMATION THEORY

## A.1 COMPUTABILITY

This definition matches Li & Vitányi (2019, Definition 1.7.4), except for the definition of a computable real number, which we adapted from Li & Vitányi (2019, Exercise 1.7.22), and for the definition of a computable set, which matches that of Li & Vitányi (2019, page 32).

**Definition A.1.** *Consider a subset $\mathcal{E}$ of $\mathbb{N}_{\geq 0}$. A map $f$ from $\mathcal{E}$ into $\mathbb{N}_{\geq 0}^3$ is said to be computable if it corresponds to a Turing machine (Li & Vitányi, 2019, Section 1.7.1). This notion extends to functions having as domain other common countable sets, such as $\mathbb{N}_{\geq 0}^k$ for $k \in \mathbb{N}$, and $\{0,1\}^*$, or any subset thereof, by identifying elements of these sets with non-negative integers via some reference bijections. Consider a computable map $f$ from a subset $\mathcal{E}$ of $\mathbb{N}_{\geq 0}$ into $\{0,1\} \times \mathbb{N}_{\geq 0} \times \mathbb{N}$. Then, composing $f$ with $(s, a, b) \mapsto (2s - 1)a/b$ yields a map from $\mathcal{E}$ to $\mathbb{Q}$, which is said to be a computable map from $\mathcal{E}$ to $\mathbb{Q}$. A map $f$ from a subset $\mathcal{E}$ of $\mathbb{N}_{\geq 0}$ into $\mathbb{R}$ is said to be lower semi-computable if there exists a computable function $\varphi$ from $\mathcal{E} \times \mathbb{N}$ into $\mathbb{Q}$, such that*

$$\forall x \in \mathcal{E}, \varphi(x, k) \underset{k \to \infty}{\to} f(x), \qquad and \qquad \forall x \in \mathcal{E}, \forall k \in \mathbb{N}, \ \varphi(x, k+1) \geq \varphi(x, k).$$

*Moreover, $f$ is said to be a computable map from $\mathcal{E}$ to $\mathbb{R}$ if both $f$ and $-f$ are lower semi-computable. A real number $\lambda$ is said to be computable if the constant function $f : \mathbb{N}_{\geq 0} \to \mathbb{R}$, $n \mapsto \lambda$ is a computable function from $\mathbb{N}_{\geq 0}$ to $\mathbb{R}$. A (possibly infinite) subset $\mathcal{X}$ of $\mathbb{N}_{\geq 0}$, is said to be computable if there exists a computable function $f$ from $\mathbb{N}_{\geq 0}$ to $\{0,1\}$, which returns $1$ if its input is in $\mathcal{X}$, and $0$ otherwise.*

The following lemma allows to construct (semi-)computable functions. Its proof is deferred to Appendix K.

**Lemma A.2.** *Let $\mathcal{E}$ denote a non-empty subset of $\mathbb{N}_{\geq 0}$, and let $f$ and $g$ denote functions from $\mathcal{E}$ to $\mathbb{R}$. (i) If $f$ and $g$ are both lower semi-computable, then functions $f + g$, $\lceil f \rceil$, and $2^f$ are lower semi-computable. If, in addition, $f$ and $g$ only take non-negative values, then $fg$ and $2^f/(3+f)^2$ are lower semi-computable. If, in addition, $f$ only takes positive values, then $\log(f)$ is lower semi-computable. (ii) If $f$ and $g$ are both computable, then functions $f + g$, $fg$, and $|f|$ are computable. If, in addition, $f$ only takes positive values, then functions $1/f$, and $f^{1/b}$ are computable, for any positive integer $b$. (iii) Let $\mathcal{X}$ be a computable finite subset of $\{0,1\}^*$. If $f$ is a lower semi-computable function from $\{0,1\}^*$ into $\mathbb{R}$, then the function $\tilde{f} : \{0,1\}^* \to \mathbb{R}$ which is null outside of $\cup_{n \in \mathbb{N}} \mathcal{X}^n$, and is defined by*

$$\forall x \in \cup_{n \in \mathbb{N}} \mathcal{X}^n, \ \tilde{f}(x) = \sum_{y \in \mathcal{X}^{l(x)}} f(y),$$

*is lower semi-computable. Moreover, if $p$ is a lower semi-computable probability measure on $\mathcal{X}$, then $p^{\otimes *}$ is lower semi-computable.*

## A.2 UNIVERSAL CRITICS AND SEMI-MEASURES

**Definition A.3.** *Given a finite set $\mathcal{W}$, a function $f : \mathcal{W} \to [0,1]$ is a semi-measure if*

$$\sum_{w \in \mathcal{W}} f(w) \leq 1.$$

*It is said to be a lower semi-computable semi-measure if $f$ is a semi-measure and $f$ is lower semi-computable.*

The following theorem, corresponds to Definition 4.3.2, Equation (4.2), and Theorems 4.3.1 and 4.3.3 in Li & Vitányi (2019). It introduces the notion of *universal $p^{\otimes *}$-critic*, used in Theis (2024). The mixture $\mathbf{m}$ therein can be used as a prior distribution, which has been shown to be relevant in machine learning applications involving realism, such as outlier detection and generative modeling (Theis, 2024).

**Theorem A.4.** *Consider a finite set $\mathcal{X}$, each element of which is identified with a string in $\{0,1\}^s$, for some $s \in \mathbb{N}$. Let $p$ be a computable distribution on $\mathcal{X}$ such that $\forall x \in \mathcal{X}, p(x) > 0$. Then, there*

*exists a $p^{\otimes*}$-critic $\delta_0$ (which is not necessarily lower semi-computable), such that for any lower semi-computable $p^{\otimes*}$-critic $\delta$, there exists a constant $c_\delta$ such that*

$$\forall x \in \bigcup_{n \in \mathbb{N}} \mathcal{X}^n, \quad \delta_0(x) \geq \delta(x) - c_\delta. \tag{23}$$

*Any $p^{\otimes*}$-critic satisfying (23) is called a universal $p^{\otimes*}$-critic.*

Since our definitions are slightly different from the classical ones, we provide a proof of Theorem A.4 in Appendix J. Such a critic $\delta_0$ is one of the best measures of realism deficiency according to $p$, in the limit of arbitrarily long strings. If a critic $\delta$ identifies a certain amount of deficiency in a given string, then $\delta_0$ will identify at least as much deficiency, up to some additive constant. Intuitively, $\delta_0$ is sensitive to all properties of randomness according to $p$. The existence of such a $\delta_0$ constitutes a remarkable property of the set of all lower semi-computable $p_X^{\otimes*}$-critics (which is infinite).

**Remark A.5.** *(Li & Vitányi, 2019, Theorem 4.3.3) The universal semi-measure $\mathbf{m}$ can be chosen in such a way that*

$$\forall x \in \{0,1\}^*, \quad |-\log(\mathbf{m}(x)) - K(x)| \leq c, \tag{24}$$

*for some constant $c$, where $K$ is the Kolmogorov complexity (Li & Vitányi, 2019, Section 3.1). Property (24) constitutes a strong result, since the Kolmogorov complexity is only defined up to a constant -we omit the corresponding details, for the purpose of clarity. The map $x \mapsto \log(1/p(x)) - K(x)$ is sometimes considered to be an approximation of a universal $p^{\otimes*}$ critic, see, e.g., Theis (2024), and Appendix J.*

# B    PROOF OF THEOREM 4.2

## B.1    OUTLINE

To show the achievability of a tuple $(R_1, \Delta_1, C_1)$, it is not necessary to construct an explicit compression scheme: it is sufficient to prove the abstract existence of such a scheme. To that end, we consider a set of random reconstructions, and study its realism properties in Section B.2. Then, we show the existence of a suitable choice of realizations of the latter reconstructions in Section B.3. In Section B.4, we prove Theorem 4.2 by proposing a compression scheme achieving a close-to-uniform sampling from the set of reconstructions. For the remainder of Section B, we fix a finite set $\mathcal{X}$ such that $|\mathcal{X}| \geq 2$, a computable distribution $p_X$ on $\mathcal{X}$ such that $\forall x \in \mathcal{X}, p_X(x) > 0$, a positive integer $B$, and a $p_X^{\otimes B}$-critic $\delta$.

## B.2    REALISM PERFORMANCE OF A UNIFORMLY SAMPLED BATCH OF RANDOM RECONSTRUCTIONS

### B.2.1    RANDOM CANDIDATE RECONSTRUCTIONS

Given a positive real $R_1$, let $\mathcal{C}$ be a family of $\lfloor 2^{R_1} \rfloor$ i.i.d. variables, each sampled from $p_X$. The $m$-th variable is denoted $y(\mathcal{C}, m)$. We denote their joint distribution by $Q_\mathcal{C}$. Given a realization $c$ of $\mathcal{C}$, we consider a batch $\mathbf{y}^{(B)}$ of $B$ elements of $c$, sampled uniformly with replacement. Then, we compute the batch's realism score $\delta(\mathbf{y}^{(B)})$. This is formalized in the following lemma, which gives an upper bound of the expected score with respect to $Q_\mathcal{C}$.

**Lemma B.1.** *Consider a positive real $R_1 \in (\log(B), \infty)$, and the following pmf.*

$$Q_{\mathcal{C}, \mathbf{M}^{(B)}, \mathbf{Y}^{(B)}} \left( \{y(m')\}_{m' \in [\lfloor 2^{R_1} \rfloor]}, \mathbf{m}^{(B)}, \mathbf{y}^{(B)} \right)$$

$$:= \left( \prod_{m'=1}^{\lfloor 2^{R_1} \rfloor} p_X(y(m')) \right) \cdot \frac{1}{\lfloor 2^{R_1} \rfloor^B} \cdot \prod_{k=1}^{B} \mathbf{1}_{y^{(k)} = y(m^{(k)})}. \tag{25}$$

*Then, we have*

$$\mathbb{E}_Q[\delta(\mathbf{Y}^{(B)})] \leq \frac{B^2}{\lfloor 2^{R_1} \rfloor} \max_x B \log \frac{1}{p_X(x)}. \tag{26}$$

The remainder of Section B.2 is dedicated to the proof of Lemma B.1. Fix $R_1 > \log(B)$.

### B.2.2 REALISM PERFORMANCE

**Claim B.2.** *Since $R_1 > \log(B)$, a simple bound yields,*

$$(p^{\mathcal{U}}_{[\lfloor 2^{R_1} \rfloor]})^{\otimes B}(M^{(1)}, ..., M^{(B)} \text{ 2 by 2 distinct}) \geq 1 - \frac{B^2}{\lfloor 2^{R_1} \rfloor}.$$

See Appendix I for a proof. From the definition (Section B.2.1) of $Q$, for any $\mathcal{E} \in \mathcal{B}(\mathbb{R})$,

$$Q\Big(\Big\{\delta_0\big(\{y(\mathcal{C}, M^{(k)})\}_{k \in [B]}\big) \in \mathcal{E}\Big\}$$
$$\Big| \Big\{M^{(1)}, ..., M^{(B)} \text{ 2 by 2 distinct}\Big\}\Big)$$
$$= p_X^{\otimes B}\Big(\delta_0\big(\mathbf{X}^{(B)}\big) \in \mathcal{E}\Big). \tag{27}$$

Therefore, we have

$$\mathbb{E}_Q[\delta(\{y(\mathcal{C}, M^{(k)})\}_{k \in [B]})]$$
$$= \sum_{\mathbf{m}^{(B)}} \mathbb{E}_Q[\mathbf{1}_{\mathbf{M}^{(B)}=\mathbf{m}^{(B)}} \delta(\{y(\mathcal{C}, m^{(k)})\}_{k \in [B]})]$$
$$= \sum_{\mathbf{m}^{(B)}} \mathbb{E}_Q[\mathbf{1}_{\mathbf{M}^{(B)}=\mathbf{m}^{(B)}}] \mathbb{E}_Q[\delta(\{y(\mathcal{C}, m^{(k)})\}_{k \in [B]})]$$
$$= \sum_{\{m^{(k)}\}_{k \in [B]} \text{ 2 by 2} \neq} (p^{\mathcal{U}}_{[\lfloor 2^{R_1} \rfloor]})^{\otimes B}(\mathbf{M}^{(B)}=\mathbf{m}^{(B)}) \mathbb{E}_{p_X^{\otimes B}}[\delta(\mathbf{X}^{(B)})]$$
$$+ \sum_{\{m^{(k)}\}_{k \in [B]} \text{ not 2 by 2} \neq} (p^{\mathcal{U}}_{[\lfloor 2^{R_1} \rfloor]})^{\otimes B}(\mathbf{M}^{(B)}=\mathbf{m}^{(B)}) \mathbb{E}_Q[\delta(\{y(\mathcal{C}, m^{(k)})\}_{k \in [B]})]$$
$$\leq \mathbb{E}_{p_X^{\otimes B}}[\delta(\mathbf{X}^{(B)})] + \max(\delta)(p^{\mathcal{U}}_{[\lfloor 2^{R_1} \rfloor]})^{\otimes B}(M^{(1)}, ..., M^{(B)} \text{not 2 by 2} \neq)$$
$$\leq \mathbb{E}_{p_X^{\otimes B}}[\delta(\mathbf{X}^{(B)})] + \frac{B^2}{\lfloor 2^{R_1} \rfloor} \max_x B \log \frac{1}{p_X(x)}, \tag{28}$$

where (28) follows from Claim B.2 and (3).

**Claim B.3.** *For any distribution $p$ on a finite set, any $p$-critic $\delta$ satisfies*

$$\mathbb{E}_p[\delta(X)] \leq 0. \tag{29}$$

*Proof.* By setting $q : \mathbf{x} \mapsto p(x) \cdot 2^{\delta(x)}$, we can write

$$\forall x \in \mathcal{X} \text{ s.t. } p(x) > 0, \ \delta(x) = \log\Big(\frac{q(x)}{p(x)}\Big), \text{ with } 0 < \sum_{x \in \mathcal{X}} q(x) \leq 1. \tag{30}$$

We denote the latter sum by $q(\mathcal{X})$. Then, $q/q(\mathcal{X})$ is a probability distribution on $\mathcal{X}$, and we have

$$\mathbb{E}_p[\delta(X)] \leq \mathbb{E}_p\Big[\log\Big(\frac{q(X)/q(\mathcal{X})}{p(X)}\Big)\mathbf{1}_{p(X)>0}\Big] = -KL(p||q/q(\mathcal{X})) \leq 0. \tag{31}$$

$\square$

This concludes the proof of Lemma B.1.

### B.3 FURTHER PROPERTIES OF A UNIFORMLY SAMPLED BATCH

**Proposition B.4.** *Consider a finite set $\mathcal{X}$ such that $|\mathcal{X}| \geq 2$, a distribution $p_X$ on $\mathcal{X}$ such that $\forall x \in \mathcal{X}, p_X(x) > 0$, a positive integer $B$, some $R > \log(B)/\log(|\mathcal{X}|)$, some $\Delta \in \mathbb{R}_+$, and a $p_X^{\otimes B}$-critic $\delta$. Consider any conditional transition kernel $p_{Y|X}$ from $\mathcal{X}$ to $\mathcal{X}$ satisfying*

$$p_Y \equiv p_X, \quad \mathbb{E}_p[d(X, Y)] \leq \Delta. \tag{32}$$

*Then, for any $\varepsilon \in (0, \Delta/2)$, and any $\gamma > 0$, there exists a family $\{y(m)\}_{m \in [\lfloor 2^{R_1} \rfloor]}$, denoted $\mathbf{c}$, of elements of $\mathcal{X}$, such that distribution*

$$Q_{M,Y,X}\left(m, y, x\right) := \frac{1}{\lfloor 2^{R_1} \rfloor} \cdot \left(\mathbf{1}_{y=y(m)}\right) \cdot p_{X|Y=y(m)}(x) \tag{33}$$

*satisfies*

$$\|Q_X - p_X\|_{TV} \leq \frac{3\Delta}{\varepsilon}[p(\mathcal{A}_{R,\gamma}) + 2^{-\gamma \log(|\mathcal{X}|)/2}] \tag{34}$$

$$\mathbb{E}_{Q^{\otimes B}}[d(\mathbf{X}^{(B)}, \mathbf{Y}^{(B)})] \leq \Delta + \varepsilon \tag{35}$$

$$\mathbb{E}_{Q^{\otimes B}}[\delta(\mathbf{Y}^{(B)})] \leq \frac{3\Delta}{\varepsilon} \cdot \frac{B^2}{\lfloor 2^{R_1} \rfloor} \max_x B \log \frac{1}{p_X(x)}, \tag{36}$$

*where $R_1 = R \log |X|$, and*

$$\mathcal{A}_{R,\gamma} := \left\{(x,y) \in \mathcal{X}^2 \mid \log\left(\frac{p_{X,Y}(x,y)}{p_X(x)p_Y(y)}\right) - \log(\lfloor 2^{R_1} \rfloor) > -\gamma \log(|\mathcal{X}|)\right\}. \tag{37}$$

*Proof.* Fix some $R > \log(B)/\log(|\mathcal{X}|)$, some $\Delta > 0$, some $\varepsilon \in (0, \Delta/2)$, some $\gamma > 0$, and a conditional transition kernel $p_{Y|X}$ from $\mathcal{X}$ to $\mathcal{X}$ satisfying

$$p_Y \equiv p_X, \quad \mathbb{E}_p[d(X,Y)] \leq \Delta. \tag{38}$$

Define $R_1 = R \log |\mathcal{X}|$. We apply Lemma B.1, and use the notation therein. Then, from Markov's inequality, we have

$$Q_{\mathcal{C}}\left(\mathbb{E}_Q[\delta(\mathbf{Y}^{(B)})|\mathcal{C}] \geq \frac{3\Delta}{\varepsilon} \frac{B^2}{\lfloor 2^{R_1} \rfloor} \max_x B \log \frac{1}{p_X(x)}\right) \leq \frac{\varepsilon}{3\Delta}. \tag{39}$$

We extend distribution $Q$ as follows.

$$Q_{\mathcal{C}, \mathbf{M}^{(B)}, \mathbf{Y}^{(B)}, \mathbf{X}^{(B)}}\left(\{y(m')\}_{m' \in [\lfloor 2^{R_1} \rfloor]}, \mathbf{m}^{(B)}, \mathbf{y}^{(B)}, \mathbf{x}^{(B)}\right) :=$$

$$Q_{\mathcal{C}, \mathbf{M}^{(B)}, \mathbf{Y}^{(B)}}\left(\{y(m')\}_{m' \in [\lfloor 2^{R_1} \rfloor]}, \mathbf{m}^{(B)}, \mathbf{y}^{(B)}\right) \cdot \prod_{k=1}^B p_{X|Y=y(m^{(k)})}(x^{(k)}). \tag{40}$$

Distribution $Q_{\mathcal{C}, M^{(1)}, Y^{(1)}, X^{(1)}}$ corresponds to the setting of Cuff (2013, Theorem VII.1), known as the soft covering lemma. Since $p_Y \equiv p_X$, the latter lemma yields that for any $\tau \in \mathbb{R}$,

$$\mathbb{E}_{\mathcal{C}}\left[\|Q_{X^{(1)}|\mathcal{C}} - p_X\|_{TV}\right] \leq p(\mathcal{A}_\tau) + 2^{\tau/2}, \tag{41}$$

where

$$\mathcal{A}_\tau := \{(x,y) \mid \log(p_{Y|X=x}(y)/p_X(y)) - \log(\lfloor 2^{R_1} \rfloor) > \tau\}. \tag{42}$$

We choose $\tau = -\gamma \log |\mathcal{X}|$. Then, $\mathcal{A}_\tau = \mathcal{A}_{R,\gamma}$, with the notation of Proposition B.4. Hence, from (41) and Markov's inequality, we have

$$Q_{\mathcal{C}}\left(\|Q_{X^{(1)}|\mathcal{C}} - p_X\|_{TV} \geq \frac{3\Delta}{\varepsilon}[p(\mathcal{A}_{R,\gamma}) + 2^{-\gamma \log(|\mathcal{X}|)/2}]\right) \leq \frac{\varepsilon}{3\Delta}. \tag{43}$$

By construction, we have $Q_{\mathbf{Y}^{(B)}, \mathbf{X}^{(B)}} \equiv p_{Y,X}^{\otimes B}$. Therefore, from (38), and the additivity of $d$, we have

$$\mathbb{E}_Q[d(\mathbf{X}^{(B)}, \mathbf{Y}^{(B)})] \leq \Delta. \tag{44}$$

Therefore, from Markov's inequality,

$$Q_{\mathcal{C}}\left(\mathbb{E}_Q[d(\mathbf{X}^{(B)}, \mathbf{Y}^{(B)})|\mathcal{C}] \geq \Delta + \varepsilon\right) \leq \frac{\Delta}{\Delta + \varepsilon} = 1 - \frac{\varepsilon}{\Delta} \cdot \frac{1}{1 + \varepsilon/\Delta} < 1 - \frac{2\varepsilon}{3\Delta}, \tag{45}$$

where we have used the fact that $\varepsilon \in (0, \Delta/2)$. From a union bound and (39), (43), and (45) there exists a realization $c_*$ of $\mathcal{C}$ such that none of the corresponding events hold. Since, by construction,

$$Q_{\mathbf{M}^{(B)}, \mathbf{Y}^{(B)}, \mathbf{X}^{(B)}|\mathcal{C}=c*} \equiv Q_{M^{(1)}, Y^{(1)}, X^{(1)}|\mathcal{C}=c*}^{\otimes B},$$

this concludes the proof of Proposition B.4. $\qquad \square$

### B.4 PROOF OF THEOREM 4.2

Fix some $R > \log(B)/\log(|\mathcal{X}|)$, some $\Delta > 0$, some $\varepsilon \in (0, \Delta/2)$, some $\gamma > 0$, and a conditional transition kernel $p_{Y|X}$ from $\mathcal{X}$ to $\mathcal{X}$ satisfying

$$p_Y \equiv p_X, \quad \mathbb{E}_p[d(X,Y)] \leq \Delta. \tag{46}$$

Define $R_1 = R \log |\mathcal{X}|$. Then, we can apply Proposition B.4. We use the notation from the latter.

#### B.4.1 COMPRESSION SCHEME ACHIEVING CLOSE-TO-UNIFORM SAMPLING

We define the following distribution $P_{X,Y,M}$, which differs from $Q$ in having the correct marginal for $X$ :

$$P_{X,M,Y} := p_X \cdot Q_{M,Y|X}. \tag{47}$$

Therefore, from (33), distribution $P$ satisfies Markov chain $X-M-Y$. Hence, it defines a $(R_1, 0)$ code. From Lemma H.2 (Appendix H), comparing $P$ with $Q$ reduces to comparing marginals, i.e. to (34) :

$$\begin{aligned}
\left\|P_{M,X,Y}-Q_{M,X,Y}\right\|_{TV} &= \left\|P_X-Q_X\right\|_{TV} \\
&= \left\|p_X - Q_X\right\|_{TV} \leq \frac{3\Delta}{\varepsilon}[p(\mathcal{A}_{R,\gamma}) + 2^{-\gamma \log(|\mathcal{X}|)/2}].
\end{aligned} \tag{48}$$

Since $d$ is additive, we have

$$\mathbb{E}_{(P)^{\otimes B}}[d(\mathbf{X}^{(B)}, \mathbf{Y}^{(B)})] = \mathbb{E}_P[d(X,Y)] \text{ and}$$

$$\mathbb{E}_{(Q)^{\otimes B}}[d(\mathbf{X}^{(B)}, \mathbf{Y}^{(B)})] = \mathbb{E}_Q[d(X,Y)].$$

Since $d$ is bounded, then we can apply Lemma H.3 (Appendix H). Then, from (48), and Lemma H.1 with $W = (X,Y)$, we have

$$\begin{aligned}
\mathbb{E}_{P^{\otimes B}}[d(\mathbf{X}^{(B)}, \mathbf{Y}^{(B)})] &\leq \mathbb{E}_{Q^{\otimes B}}[d(\mathbf{X}^{(B)}, \mathbf{Y}^{(B)})] + \frac{6\Delta}{\varepsilon} \max(d)[p(\mathcal{A}_{R,\gamma}) + 2^{-\gamma \log(|\mathcal{X}|)/2}] \\
&\leq \Delta + \varepsilon + \frac{6\Delta}{\varepsilon} \max(d)[p(\mathcal{A}_{R,\gamma}) + 2^{-\gamma \log(|\mathcal{X}|)/2}],
\end{aligned} \tag{49}$$

where the last inequality follows from (35). Moving to the realism performance, we have the following property of the TVD - see Appendix H:

**Claim B.5.** *Given any two distributions $P$ and $Q$ on the same finite alphabet, we have, for any $B \in \mathbb{N}$,*

$$\left\|P^{\otimes B}-Q^{\otimes B}\right\|_{TV} \leq B\left\|P-Q\right\|_{TV}.$$

From Lemma H.3, Claim B.5, (48), and Lemma H.1 with $W = \mathbf{Y}^{(B)}$, we have,

$$\mathbb{E}_{P^{\otimes B}}[\delta(\mathbf{Y}^{(B)})] \leq \mathbb{E}_{Q^{\otimes B}}[\delta(\mathbf{Y}^{(B)})] + \frac{6B\Delta}{\varepsilon} \max(\delta)[p(\mathcal{A}_{R,\gamma}) + 2^{-\gamma \log(|\mathcal{X}|)/2}]$$

$$\leq \frac{3\Delta}{\varepsilon} \cdot \frac{B^2}{\lfloor 2^{R_1} \rfloor} \max_x B \log \frac{1}{p_X(x)} + \frac{6B\Delta}{\varepsilon}[p(\mathcal{A}_{R,\gamma}) + 2^{-\gamma \log(|\mathcal{X}|)/2}] \cdot \max_x B \log \frac{1}{p_X(x)}. \tag{50}$$

This concludes the proof.

## C ACHIEVABILITY OF THEOREM 4.1

Consider some $\Delta \in \mathbb{R}_+$ such that $R^{(1)}(\Delta, 0) < H_p(X)$, and a sequence $\{B_n\}_{n \geq 1}$ of positive integers such that

$$\log(B_n)/n \underset{n \to \infty}{\longrightarrow} 0. \tag{51}$$

Fix $R \in (R^{(1)}(\Delta, 0), H_p(X))$, $\varepsilon \in (0, R - R^{(1)}(\Delta, 0))$, and $\gamma \in (0, \varepsilon/\log(|\mathcal{X}|))$. Then, there exists $p_{Y|X}$ such that

$$p_Y \equiv p_X, \ \mathbb{E}_p[d(X,Y)] \leq \Delta, \ R \geq I_p(X;Y) + \varepsilon. \tag{52}$$

We use the powerful result of Theorem A.4 regarding the existence of a so-called *universal critic*. From Definition 2.1, for every $n \in \mathbb{N}$, the restriction of $\delta_0$ to $\mathcal{X}^{nB_n}$ is a $p_X^{\otimes nB_n}$-critic. Moreover, from (51), for large enough $n$, we have $nR > \log(B_n)$. Then, for large enough $n$, we can apply Theorem 4.2 for set $\mathcal{X}^n$, distribution $p_X^{\otimes n}$, transition kernel $\prod_{t=1}^n p_Y|X$, batch size $B_n$, critic $\delta_0$, rate $nR/\log(|\mathcal{X}^n|)$, and constants $\Delta, \varepsilon, \gamma$. This gives that, for every $n$ large enough, there is a $(n, R, 0)$ code, inducing a distribution $P^{(n)}$ such that

$$\mathbb{E}_{(P^{(n)})\otimes B_n}\left[d(\mathbf{X}^{(n,B_n)}, \mathbf{Y}^{(n,B_n)})\right] \leq \Delta + \varepsilon + \frac{6\Delta}{\varepsilon} \max(d)[p(\mathcal{A}_{R,\gamma}^{(n)}) + 2^{-\gamma n \log(|\mathcal{X}|)/2}], \quad (53)$$

$$\mathbb{E}_{(P^{(n)})\otimes B_n}\left[\delta_0(\mathbf{Y}^{(n,B_n)})\right] \leq$$
$$\frac{3\Delta}{\varepsilon}\left[\frac{B_n^2}{\lfloor 2^{nR}\rfloor} \max_x nB_n \log\frac{1}{p_X(x)} + 2B_n[p(\mathcal{A}_{R,\gamma}^{(n)}) + 2^{-\gamma n \log(|\mathcal{X}|)/2} \cdot \max_x nB_n \log\frac{1}{p_X(x)}]\right], \quad (54)$$

where

$$\mathcal{A}_{R,\gamma}^{(n)} := \left\{(x_{1:n}, y_{1:n}) \in (\mathcal{X}^n)^2 \mid \sum_{t=1}^n \log\left(\frac{p_{X,Y}(x_t, y_t)}{p_X(x_t)p_Y(y_t)}\right) - \log(\lfloor 2^{nR}\rfloor) > -\gamma n \log(|\mathcal{X}|)\right\}, \quad (55)$$

with the convention $0/0 := 1$. From (52), $\log(\lfloor 2^{nR}\rfloor)/n - \gamma \log(|\mathcal{X}|) > I_p(X;Y)$ for large enough $n$. Then, since $\mathcal{X}$ is finite, we have, from Hoeffding's inequality,

$$p(\mathcal{A}_{R,\gamma}^{(n)}) = O(e^{-\kappa n}), \quad (56)$$

for some $\kappa > 0$. Hence, from (51), (53), (54), and Theorem A.4, we have

$$\limsup_{n\to\infty} \mathbb{E}_{(P^{(n)})\otimes B_n}\left[d(\mathbf{X}^{(n,B_n)}, \mathbf{Y}^{(n,B_n)})\right] \leq \Delta + \varepsilon, \quad (57)$$

and for any lower semi-computable $p_X^{\otimes *}$-critic $\delta$,

$$\sup_{n\in\mathbb{N}} \mathbb{E}_{(P^{(n)})\otimes B_n}\left[\delta(\mathbf{Y}^{(n,B_n)})\right] < \infty. \quad (58)$$

From the proof of Theorem 4.2, we know that $P^{(n)}$ has a deterministic decoder. Hence, it only remains to derandomize the encoder of $P^{(n)}$. We denote its decoder by $m \mapsto y_{1:n}(m)$. The following claim is a slight modification of Hamdi et al. (2024, Proposition 4). We provide details in Section C.1.

**Claim C.1.** *There exists a sequence of deterministic maps*

$$f^{(n)} : \mathcal{X}^n \to [2^{nR}], \quad \text{such that}$$
$$\left\|\hat{\tilde{P}}_{\mathcal{X}^2}^{(n)}[X^n, y_{1:n}(M)] - \hat{P}_{\mathcal{X}^2}^{(n)}[X^n, y_{1:n}(M)]\right\|_{TV} \xrightarrow[n\to\infty]{} 0,$$
$$\liminf_{n\to\infty} \frac{-1}{n}\log\left\|\tilde{P}_M^{(n)} - P_M^{(n)}\right\|_{TV} > 0, \quad \text{where} \qquad (59)$$
$$\tilde{P}_{X^n, M}^{(n)} := p_X^{\otimes n} \cdot \mathbf{1}_{M=f^{(n)}(X^n)}.$$

Then, from (51) and Claim B.5, we have

$$\liminf_{n\to\infty} \frac{-1}{n}\log\left\|(\tilde{P}^{(n)})_M^{\otimes B_n} - (P^{(n)})_M^{\otimes B_n}\right\|_{TV} > 0. \quad (60)$$

Thus, from Lemma H.3 and (3), we have

$$\left|\mathbb{E}_{(\tilde{P}^{(n)})\otimes B_n}\left[\delta(\mathbf{Y}^{(n,B_n)})\right] - \mathbb{E}_{(P^{(n)})\otimes B_n}\left[\delta(\mathbf{Y}^{(n,B_n)})\right]\right| \xrightarrow[n\to\infty]{} 0. \quad (61)$$

Moreover, since $d$ is bounded, then from Lemma H.3, we obtain

$$\left|\mathbb{E}_{(P^{(n)})\otimes B_n}\left[d(\mathbf{X}^{(n,B_n)}, \mathbf{Y}^{(n,B_n)})\right] - \mathbb{E}_{(P^{(n)})\otimes B_n}\left[d(\mathbf{X}^{(n,B_n)}, \mathbf{Y}^{(n,B_n)})\right]\right| \xrightarrow[n\to\infty]{} 0. \quad (62)$$

Since this analysis is valid for any $\varepsilon \in (0, R - R^{(1)}(\Delta, 0))$, then tuple $(R, \{B_n\}_{n\geq 1}, \Delta)$ is asymptotically achievable with algorithmic realism with fully deterministic codes. This being true for any $R \in (R^{(1)}(\Delta, 0), H_p(X))$, we have

$$R(\Delta) \leq R_*(\Delta) \leq R^{(1)}(\Delta, 0),$$

as desired.

## C.1 ENCODER DERANDOMIZATION

We show that Claim C.1 follows from Hamdi et al. (2024, Proposition 4), and its proof. We can apply that result directly, since $R < H_p(X)$ and $\mathcal{X}$ is finite. This would give all properties in Claim C.1, except for the exponential decay in (59). To obtain the latter, it is sufficient to adapt the proof of Hamdi et al. (2024, Proposition 4), by replacing the use of the law of large numbers with the use of Hoeffding's inequality, and using Cuff (2013, Theorem VII.1) with $\tau = -n\gamma$, for small enough $\gamma$.

# D CONVERSE OF THEOREM 4.1

From standard information-theoretic arguments, we have the following result - see Appendix F for a proof.

**Lemma D.1.** *Consider a triplet* $(R, R_c, \Delta)$ *and a sequence of codes, the $n$-th being* $(n, R, R_c)$, *inducing a sequence* $\{P^{(n)}_{X_{1:n}, J, M, Y_{1:n}}\}_{n \geq 1}$ *of distributions such that*

$$\limsup_{n \to \infty} \mathbb{E}_{(P^{(n)})^{\otimes b_n}}\left[d(\mathbf{X}^{(n,b_n)}, \mathbf{Y}^{(n,b_n)})\right] \leq \Delta, \tag{63}$$

*for some sequence* $\{b_n\}_{n \geq 1}$ *of positive integers. For every $n \geq 1$, let $T^{(n)}$ denote a uniform variable on $[nb_n]$ independent from all other random variables. Then, there exists a conditional distribution $p_{Y|X}$ and an increasing sequence $\{n_i\}_{i \geq 1}$ of positive integers such that*

$$(P^{(n_i)})^{\otimes b_{n_i}}_{X_{T^{(n_i)}}, Y_{T^{(n_i)}}} \xrightarrow[i \to \infty]{} p_{X,Y} \tag{64}$$

$$\Delta \geq \mathbb{E}_p[d(X, Y)] \tag{65}$$

$$R \geq I_p(X; Y), \tag{66}$$

*where $p_{X,Y}$ refers to $p_X \cdot p_{Y|X}$.*

## D.1 CONVERSE PROOF

Consider some $\Delta \in \mathbb{R}_+$ such that $R^{(1)}(\Delta, 0) < H_p(X)$, and a sequence $\{B_n\}_{n \geq 1}$ of positive integers such that

$$\log(B_n)/n \xrightarrow[n \to \infty]{} 0. \tag{67}$$

We know that $R_*(\Delta) \geq R(\Delta)$, and prove that $R(\Delta) \geq R^{(1)}(\Delta, 0)$. Consider a couple $(R, \Delta) \in \mathbb{R}^2_+$, and some $R_c \in \mathbb{R}_{\geq 0}$ such that $(R, R_c, \{B_n\}_{n \geq 1}, \Delta)$ is asymptotically achievable with algorithmic realism. Fix $\varepsilon > 0$. Then, there exists a sequence of codes, the $n$-th being $(n, R, R_c)$, inducing a sequence $\{P^{(n)}_{X_{1:n}, J, M, Y_{1:n}}\}_n$ of distributions such that

$$\limsup_{n \to \infty} \mathbb{E}_{(P^{(n)})^{\otimes B_n}}\left[d(\mathbf{X}^{(n,B_n)}, \mathbf{Y}^{(n,B_n)})\right] \leq \Delta + \varepsilon, \tag{68}$$

and for any lower semi-computable $p_X^{\otimes *}$-critic $\delta$,

$$\sup_{n \in \mathbb{N}} \mathbb{E}_{(P^{(n)})^{\otimes B_n}}\left[\delta(\mathbf{Y}^{(n,B_n)})\right] < \infty. \tag{69}$$

Then, Lemma D.1 applies, with $b_n = B_n$, for all $n$, with $R + \varepsilon$ instead of $R$, and $\Delta + \varepsilon$ instead of $\Delta$. Then, there exists a conditional distribution $p_{Y|X}$ and an increasing sequence $\{n_i\}_{i \geq 1}$ of positive integers such that

$$(P^{(n_i)})^{\otimes b_{n_i}}_{X_{T^{(n_i)}}, Y_{T^{(n_i)}}} \xrightarrow[i \to \infty]{} p_{X,Y} \tag{70}$$

$$\Delta + \varepsilon \geq \mathbb{E}_p[d(X, Y)] \tag{71}$$

$$R + \varepsilon \geq I_p(X; Y), \tag{72}$$

where for any $n \in \mathbb{N}$, variable $T^{(n)}$ is uniformly distributed on $[nB_n]$, and independent from all other random variables. We prove that $p_Y \equiv p_X$. Fix $e_0 \in \mathcal{X}$. Consider the computable $p_X^{\otimes *}$-critic $\delta$ from Claim G.1, with $q$ therein taken to be $p_X$. Then, from (69),

$$\sup_{n \in \mathbb{N}} \mathbb{E}_{(P^{(n)})^{\otimes B_n}}\left[\delta(\mathbf{Y}^{(n,B_n)}) - 2\log(\delta(\mathbf{Y}^{(n,B_n)}) + 3)\right] < \infty. \tag{73}$$

Thus,

$$\sup_{n \in \mathbb{N}} \mathbb{E}_{(P^{(n)})^{\otimes B_n}}\left[\delta(\mathbf{Y}^{(n,B_n)})\right] < \infty, \text{ and } \mathbb{E}_{(P^{(n)})^{\otimes B_n}}\left[\delta(\mathbf{Y}^{(n,B_n)}) - \frac{1}{2}\log(nB_n)\right] \underset{n \to \infty}{\longrightarrow} -\infty.$$

Thus, the frequency of $e_0$ in a batch of reconstructions converges in $L_1$ norm to $p_X(e_0)$. Hence, the expected frequencies converge to $p_X(e_0)$. This rewrites as

$$(P^{(n)})^{\otimes B_n}_{Y_{T^{(n)}}}(e_0) \to p_X(e_0). \tag{74}$$

This is true for any $e_0$ in $\mathcal{X}$. Thus, from (70), $p_Y \equiv p_X$. Hence, from (71) and (72), we have

$$R + \varepsilon \geq R^{(1)}(\Delta + \varepsilon, 0). \tag{75}$$

This being true for any $\varepsilon > 0$, and since $R^{(1)}(\cdot, 0)$ is convex -thus continuous- on $(0, \infty)$, we have

$$R \geq R^{(1)}(\Delta, 0). \tag{76}$$

This being true for any $R \in \mathbb{R}_+$ such that there exists $R_c \in \mathbb{R}_{\geq 0}$ such that $(R, R_c, \{B_n\}_{n \geq 1}, \Delta)$ is asymptotically achievable with algorithmic realism, we have

$$R(\Delta) \geq R^{(1)}(\Delta, 0), \tag{77}$$

as desired.

# E    PROOF OF THEOREM 4.4

Consider an increasing sequence $\{B_n\}_{n \geq 1}$ of positive integers such that

$$\frac{B_n}{|\mathcal{X}|^n} \to \infty, \tag{78}$$

some $R_c \in \mathbb{R}_{\geq 0}$, and some $(R, \Delta) \in (\mathbb{R}_+)^2$ such that tuple $(R, R_c, \Delta)$ is asymptotically achievable with near-perfect realism. From Theorem 1 in Wagner (2022), $(R, R_c, \Delta)$ achievable with *perfect realism*, i.e. satisfying the properties in Definition 4.3, with (20) replaced with

$$\exists N \in \mathbb{N}, \forall n \geq N, \ P^{(n)}_{Y_{1:n}} \equiv p_X^{\otimes n}. \tag{79}$$

Fix $\varepsilon > 0$, and a corresponding sequence of $(n, R + \varepsilon, R_c)$ codes. Denote by $P^{(n)}$ the distribution induces by the $n$-th code. Then, there exists an integer $N_\varepsilon$ such that

$$\limsup_{n \to \infty} \mathbb{E}_{P^{(n)}}\left[d(X_{1:n}, Y_{1:n})\right] \leq \Delta + \varepsilon, \tag{80}$$

$$\forall n \geq N_\varepsilon, \ (P^{(n)}_{Y_{1:n}})^{\otimes B_n} \equiv p_X^{\otimes nB_n}. \tag{81}$$

From (80), (81), Claim B.3, and the additivity of the distortion measure $d$, we have

$$\limsup_{n \to \infty} \mathbb{E}_{(P^{(n)})^{\otimes B_n}}\left[d(\mathbf{X}^{(n,B_n)}, \mathbf{Y}^{(n,B_n)})\right] \leq \Delta + \varepsilon, \tag{82}$$

and for any lower semi-computable $p_X^{\otimes *}$-critic $\delta$,

$$\sup_{n \in \mathbb{N}} \mathbb{E}_{(P^{(n)})^{\otimes B_n}}\left[\delta(\mathbf{Y}^{(n,B_n)})\right] < \infty. \tag{83}$$

Since this analysis is valid for every $\varepsilon > 0$, then $(R, R_c, \{B_n\}_{n \geq 1}, \Delta)$ is asymptotically achievable with algorithmic realism. Moving to the converse, consider a computable increasing sequence $\{B_n\}_{n \geq 1}$ of positive integers such that

$$\frac{B_n}{|\mathcal{X}|^n} \to \infty, \tag{84}$$

some $R_c \in \mathbb{R}_{\geq 0}$, and some $(R, \Delta) \in (\mathbb{R}_+)^2$ such that tuple $(R, R_c, \Delta)$ is asymptotically achievable with algorithmic realism. Fix $\varepsilon > 0$. Then, there exists a sequence of codes, the $n$-th being $(n, R + \varepsilon, R_c)$, such that the sequence $\{P^{(n)}\}_n$ of distributions induced by the codes satisfies

$$\limsup_{n \to \infty} \mathbb{E}_{(P^{(n)})^{\otimes B_n}}\left[d(\mathbf{X}^{(n,B_n)}, \mathbf{Y}^{(n,B_n)})\right] \leq \Delta + \varepsilon, \text{ and} \tag{85}$$

and for any lower semi-computable $p_X^{\otimes *}$-critic $\delta$,

$$\sup_{n \in \mathbb{N}} \mathbb{E}_{(P^{(n)})^{\otimes B_n}}\left[\delta(\mathbf{Y}^{(n,B_n)})\right] < \infty. \tag{86}$$

**Lemma E.1.** *(Canonne, 2020) There exists a positive integer $\lambda$ such that for any $k \in \mathbb{N}$, any distribution q on some finite set $\mathcal{W}$ of size $k$, any $\varepsilon, \eta > 0$, and any integer b satisfying*

$$b \geq \lambda \cdot \frac{k + \log(1/\eta)}{\varepsilon^2}, \tag{87}$$

*we have*

$$q^{\otimes b}\left(\left\|\mathbb{P}_{\mathcal{W}}^{emp}[W^b] - q\right\|_{TV} \geq \varepsilon\right) \leq \eta. \tag{88}$$

For every $n \in \mathbb{N}$, define

$$C_n := \left\lceil \left(\frac{B_n}{|\mathcal{X}|^n}\right)^{\frac{1}{3}} \right\rceil. \tag{89}$$

Since $\mathcal{X}$ is finite, $\{C_n\}_{n \geq 1}$ is a computable sequence of positive integers. Moreover, from (84), we have

$$C_n \underset{n \to \infty}{\longrightarrow} \infty. \tag{90}$$

Choosing, for every $n \in \mathbb{N}$, $\eta = 1/3$ and $\varepsilon = 1/C_n$, then from Lemma E.1 and (90) we have, for large enough $n$,

$$(P^{(n)})^{\otimes B_n}\left(\left\|\mathbb{P}_{\mathcal{X}^n}^{emp}[\mathbf{Y}^{(n,B_n)}] - P_{Y_{1:n}}^{(n)}\right\|_{TV} \geq \frac{1}{C_n}\right) \leq \frac{1}{3}. \tag{91}$$

Consider the computable sequence of positive integers defined by

$$\forall n \in \mathbb{N}, \; A_n := \left\lceil \left(\frac{B_n}{|\mathcal{X}|^n}\right)^{\frac{4}{9}} \right\rceil. \tag{92}$$

Since $\{B_n\}_{n \geq 1}$ is increasing, then for any $t \in \mathbb{N}$, there exists a unique integer $n \in \mathbb{N}_{\geq 0}$ such that

$$t \in [nB_n, (n+1)B_{n+1}),$$

with the definition $B_0 := 0$. We define $\delta : \cup_{t \in \mathbb{N}} \mathcal{X}^t \to \mathbb{N}_{\geq 0}$ as follows. For any integer $t \in [1, B_1)$, and any $x_{1:t} \in \mathcal{X}^t$, let $\delta(x) := 0$. For any $n \in \mathbb{N}$, any $t \in [nB_n, (n+1)B_{n+1})$, and any $x_{1:t} \in \mathcal{X}^t$, let

$$\delta(x_{1:t}) := \left\lceil A_n \left\|\mathbb{P}_{\mathcal{X}^n}^{emp}[x_{1:nB_n}] - p_X^{\otimes n}\right\|_{TV} \right\rceil. \tag{93}$$

**Claim E.2.** *From Lemma E.1 and (90), there exists a positive integer $L$ such that $\delta - 2\log(\delta + 3) - L$ is a lower semi-computable $p_X^{\otimes *}$-critic.*

We provide a proof in Appendix G.2. Then, we can apply (86) to critic $\delta - 2\log(\delta + 3) - L$, and get,

$$\sup_{n \in \mathbb{N}} \mathbb{E}_{(P^{(n)})^{\otimes B_n}}\left[\delta(\mathbf{Y}^{(n,B_n)}) - 2\log(\delta(\mathbf{Y}^{(n,B_n)}) + 3) - L\right] < \infty. \tag{94}$$

Thus,

$$\sup_{n \in \mathbb{N}} \mathbb{E}_{(P^{(n)})^{\otimes B_n}}\left[\delta(\mathbf{Y}^{(n,B_n)})\right] < \infty, \; \text{ and } \; (P^{(n)})^{\otimes B_n}\left(\delta(\mathbf{Y}^{(n,B_n)}) \geq C_n\right) \underset{n \to \infty}{\longrightarrow} 0,$$

because $\{C_n\}_{n \geq 1}$ tends to infinity. Combining this with (91) through a union bound, we obtain, from the triangle inequality for the TVD,

$$(P^{(n)})^{\otimes B_n}\left(\left\|P_{Y_{1:n}}^{(n)} - p_X^{\otimes n}\right\|_{TV} \leq \frac{C_n}{A_n} + \frac{1}{C_n}\right) > 0,$$

for large enough $n$. The above event does not depend on the random batch, hence the corresponding inequality is true, for large enough $n$. Since $\{C_n\}_{n \geq 1}$ tends to infinity and since from (84), (89), and (92), we have $C_n/A_n \to 0$, then we obtain

$$\left\|P_{Y_{1:n}}^{(n)} - p_X^{\otimes n}\right\|_{TV} \underset{n \to \infty}{\longrightarrow} 0. \tag{95}$$

Hence, from (85) and the additivity of $d$, we have that $(R, R_c, \Delta)$ is asymptotically achievable with near-perfect realism. This concludes the proof.

## F STANDARD CONVERSE ARGUMENTS

Here, we provide a proof of Lemma D.1 (Appendix D). The sequence of distributions $(P^{(n)})_{X_T,Y_T}^{\otimes b_n}$ can be seen as a bounded sequence in $\mathbb{R}^{2^{2s}}$, thus it admits a converging subsequence:

$$(P^{(n_i)})_{X_T,Y_T}^{\otimes b_n} \underset{i \to \infty}{\longrightarrow} p_{X,Y}. \tag{96}$$

Since $d$ is bounded, we have

$$\mathbb{E}_{(P^{(n_i)})^{\otimes b_n}}[d(X_T,Y_T)] \underset{i \to \infty}{\longrightarrow} \mathbb{E}_p[d(X,Y)]. \tag{97}$$

Since $d$ is additive, we have, for any $n \in \mathbb{N}$,

$$\mathbb{E}_{(P^{(n)})^{\otimes b_n}}\big[d(\mathbf{X}^{(n,b_n)}, \mathbf{Y}^{(n,b_n)})\big]$$
$$= \mathbb{E}_{(P^{(n)})^{\otimes b_n}}\big[d(X_T,Y_T)\big]. \tag{98}$$

From (63), (97) and (98), we have $\Delta \geq \mathbb{E}_p[d(X,Y)]$. Secondly, distribution $P^{(n)}$ satisfies

$$\begin{aligned}
nb_n R &\geq H(\{m^{(k)}\}_{k\in[b_n]} | \{J^{(k)}\}_{k\in[b_n]}) \\
&\geq I(\{m^{(k)}\}_{k\in[b_n]}; \mathbf{X}^{(n,b_n)} | \{J^{(k)}\}_{k\in[b_n]}) \\
&= I(\{m^{(k)}\}_{k\in[b_n]}, \{J^{(k)}\}_{k\in[b_n]}; \mathbf{X}^{(n,b_n)}) \\
&\geq I(\mathbf{Y}^{(n,b_n)}; \mathbf{X}^{(n,b_n)}) \\
&\geq \sum_{k=1}^{b_n} \sum_{t=1}^{n} I(Y_t^{(k)}; X_t^{(k)}) \\
&= nb_n I(Y_T; X_T | T) \\
&= nb_n I(T, Y_T; X_T) \\
&\geq nb_n I(Y_T; X_T).
\end{aligned}$$

Therefore, from (96), and by continuity of mutual information on the set of distributions on $(\{0,1\}^s)^2$, we have $R \geq I_p(X;Y)$.

## G FREQUENCY CRITICS

### G.1 CRITIC INVOLVING THE FREQUENCY OF A SPECIFIC PATTERN

The following claim, and its proof, are inspired from Li & Vitányi (2019, Lemma 4.3.5 & Exercise 2.4.1).

**Claim G.1.** *Consider a finite set $\mathcal{X}$, identified with a subset of $\{0,1\}^s$. Let $q$ be a distribution on $\mathcal{X}$ such that $\forall x \in \mathcal{X}, q(x) > 0$. Let $e_0$ be any string in $\mathcal{X}$, considered as a pattern of interest. For any $n \in \mathbb{N}$ and any $x_{1:n} \in \mathcal{X}^n$, let $S(x_{1:n})$ denote the number of occurrences of $e_0$ in $x_{1:n}$. Define map $\delta: \cup_{n\in\mathbb{N}} \to \mathbb{N}_{\geq 0}$ by*

$$\forall n \in \mathbb{N}, \forall x_{1:n} \in \mathcal{X}^n, \quad x_{1:n} \mapsto \left\lceil \log \left\lceil |S(x_{1:n}) - q(e_0)n| \, / \, \sqrt{n} \right\rceil \right\rceil. \tag{99}$$

*Then, $\delta - 2\log(\delta + 3)$ is a computable $q^{\otimes *}$-critic.*

*Proof.* From Lemma A.2, $\delta$ is lower semi-computable. Since $\delta - 2\log(\delta + 3) = \log(2^\delta / (\delta + 3)^2)$, then by Lemma A.2, $\delta - 2\log(\delta + 3)$ is lower semi-computable. For any $(n,C) \in \mathbb{N}^2$, and any $x_{1:n} \in \mathcal{X}^n$, we have:

$$\begin{aligned}
\{\delta(x_{1:n}) \geq C\} &= \left\{ \left\lceil \log \left\lceil |S(x_{1:n}) - nq(e_0)| \, / \, \sqrt{n} \right\rceil \right\rceil \geq C \right\} \\
&= \left\{ \log \left\lceil |S(x_{1:n}) - nq(e_0)| \, / \, \sqrt{n} \right\rceil > C - 1 \right\} \\
&= \left\{ \left\lceil |S(x_{1:n}) - nq(e_0)| \, / \, \sqrt{n} \right\rceil > 2^{C-1} \right\} \\
&= \left\{ |S(x_{1:n}) - nq(e_0)| \, / \, \sqrt{n} > 2^{C-1} \right\}.
\end{aligned}$$

From this and Chebyshev inequality, we obtain:

$$q^{\otimes n}(\delta(X_{1:n}) \geq C) \leq \mathbb{E}_{q^{\otimes n}}\left[\left(S(X_{1:n}) - nq(e_0)\right)^2 / n\right] / 4^{C-1}$$
$$= (q(e_0) - q(e_0)^2)/4^{C-1} \tag{100}$$
$$\leq 4^{-C}$$
$$\leq 2^{-C},$$

where (100) comes from the fact that $S(X_{1:n})$ follows a binomial distribution $B(n, q(e_0))$. Thus,

$$\mathbb{E}_{q^{\otimes n}}[\mathbf{1}_{\delta(X_{1:n})=C}] \leq 2^{-C}. \tag{101}$$

Therefore,

$$\mathbb{E}_{q^{\otimes n}}[\mathbf{1}_{\delta(X_{1:n})=C} \cdot 2^{\delta(X_{1:n}) - 2\log(\delta(X_{1:n})+3)}] \leq \frac{1}{(C+3)^2}. \tag{102}$$

This also holds for $C = 0$. Summing over $C \in \mathbb{N}_{\geq 0}$ gives, for any $n \in \mathbb{N}$,

$$\sum_{x_{1:n} \in \mathcal{X}^n} q^{\otimes n}(x_{1:n}) \cdot 2^{\delta(x_{1:n}) - 2\log(\delta(x_{1:n})+1) - 1} \leq 1. \tag{103}$$

Hence, we have that $\delta - 2\log(\delta + 3)$ is a lower semi-computable $q^{\otimes *}$-critic. $\qquad\square$

### G.2 CRITIC INVOLVING AN EMPIRICAL DISTRIBUTION

We provide a proof of Claim E.2.

**Claim G.2.** *The map* $f : \cup_{t \in \mathbb{N}} \mathcal{X}^t \to \mathbb{R}$ *defined by* $\forall t \in [1, B_1) \cap \mathbb{N}, \forall x_{1:t} \in \mathcal{X}^t, f(x_{1:t}) := 0$, *and*

$$\forall n \in \mathbb{N}, \forall t \in [nB_n, (n+1)B_{n+1}) \cap \mathbb{N}, \forall x_{1:t} \in \mathcal{X}^t, \ f(x_{1:t}) := \left\|\mathbb{P}_{\mathcal{X}^n}^{emp}[x_{1:nB_n}] - p^{\otimes n}\right\|_{TV} \tag{104}$$

*is computable.*

*Proof.* Since there exists $s \in \mathbb{N}$ such that $\mathcal{X} \subseteq \{0,1\}^s$, then, given some $x \in \cup_{t \in \mathbb{N}} \mathcal{X}^t \to \mathbb{R}$, one can compute the unique corresponding $t$ via a Turing machine. Moreover, since $\{B_n\}_{n \geq 1}$ is computable, one can further compute the unique $n$ such that $t \in [nB_n, (n+1)B_{n+1})$ via a Turing machine, as well as the empirical probability appearing in (104). For any $k \in \mathbb{N}$, and any $x_0 \in \mathcal{X}$, one can call the rational-valued computable upper and lower approximations of $p$ at point $(x_0, k)$. Then, one can go over all $y_{1:n} \in \mathcal{X}^n$, and use the explicit constructions from the proof of Lemma A.2 regarding the product, sum, and absolute value, yielding rational-valued computable upper and lower approximations of $f$. $\qquad\square$

We know that $A_{n \, n \geq 1}$ is computable. From Lemma A.2, the product of two computable functions is computable, thus lower semi-computable, and the ceiling function preserves semi-computability. Therefore, $\delta$ is lower semi-computable. Since $\delta - 2\log(\delta + 3) = \log(2^\delta/(\delta+3)^2)$, then by Lemma A.2, for any positive integer $L$, function $\delta - 2\log(\delta + 3) - L$ is lower semi-computable. It remains to prove that a certain choice of $L$ yields a $p_X^{\otimes *}$-critic. From (84) and (92), there exists $N_0 \in \mathbb{N}$ such that

$$\forall n \geq N_0, \ B_n \geq \lambda(|\mathcal{X}|^n + 2)A_n^2. \tag{105}$$

For any $n \geq N_0$, any $C \geq 2$, any integer $t \in [nB_n, (n+1)B_{n+1})$, and any $x_{1:t} \in \mathcal{X}^t$, we have:

$$\{\delta(x_{1:t}) \geq C\} = \left\{\left\lceil A_n \left\|\mathbb{P}_{\mathcal{X}^n}^{emp}[x_{1:nB_n}] - p^{\otimes n}\right\|_{TV}\right\rceil \geq C\right\}$$
$$= \left\{A_n \left\|\mathbb{P}_{\mathcal{X}^n}^{emp}[x_{1:nB_n}] - p^{\otimes n}\right\|_{TV} > C - 1\right\}.$$

From this, (105), and Lemma E.1, with distribution $p_X^{\otimes n}$, and parameters $b = B_n$, $\varepsilon = (C-1)/A_n, \eta = 2^{-C}$ we obtain,

$$\forall t \geq [N_0 B_{N_0}, \infty) \cap \mathbb{N}, \forall C \in \mathbb{N}_{\geq 2}, \quad p_X^{\otimes t}(\delta(X_{1:t}) \geq C) \leq 2^{-C}.$$

Therefore,

$$\forall t \geq [N_0 B_{N_0}, \infty) \cap \mathbb{N}, \forall C \in \mathbb{N}_{\geq 2}, \quad \mathbb{E}_{p_X^{\otimes t}}[\mathbf{1}_{\delta(X_{1:t})=C} \cdot 2^{\delta(X_{1:t}) - 2\log(\delta(X_{1:t})+3)}] \leq \frac{1}{(C+3)^2}.$$

This also holds for $C \in \{0, 1\}$. Summing over $C \in \mathbb{N}_{\geq 0}$ gives,

$$\forall t \geq [N_0 B_{N_0}, \infty) \cap \mathbb{N}, \quad \sum_{x_{1:t} \in \mathcal{X}^t} p_X^{\otimes t}(x_{1:t}) \cdot 2^{\delta(x_{1:t}) - 2\log(\delta(x_{1:t}) + 3)} \leq 1. \tag{106}$$

In order to extend this to all positive integers $t$, it is sufficient to multiply by $2^{-L}$ for some $L$ large enough. Therefore, there exists $L \in \mathbb{N}$ such that $\delta - 2\log(\delta + 3) - L$ is a lower semi-computable $p_X^{\otimes *}$-critic. This concludes the proof.

# H  ON THE TOTAL VARIATION DISTANCE

## H.1  SOME LEMMAS

**Lemma H.1.** *Let $\Pi$ and $\Gamma$ be two distributions on a set $\mathcal{W} \times \mathcal{L}$. Then*

$$\|\Pi_W - \Gamma_W\|_{TV} \leq \|\Pi_{W,L} - \Gamma_{W,L}\|_{TV}.$$

**Lemma H.2.** *Let $\Pi$ and $\Gamma$ be two distributions on a set $\mathcal{W} \times \mathcal{L}$. Then when using the same conditional probability kernel $\Pi_{L|W}$, we have*

$$\|\Pi_W \Pi_{L|W} - \Gamma_W \Pi_{L|W}\|_{TV} = \|\Pi_W - \Gamma_W\|_{TV}.$$

**Lemma H.3.** *Let $\Pi$ and $\Gamma$ be two distributions on a set $\mathcal{W}$, and $f : \mathcal{W} \to \mathbb{R}$ be a bounded function. Then,*

$$|\mathbb{E}_\Pi[f] - \mathbb{E}_\Gamma[f]| \leq 2\max|f| \cdot \|\Pi - \Gamma\|_{TV}.$$

## H.2  PROOF OF CLAIM B.5

Let $P$ and $Q$ be any two distributions on the same alphabet. Fix a positive integer $B$. Then, we have, with the convention $\Pi \otimes \Gamma^{\otimes 0} \equiv \Pi$,

$$\|P^B - Q^B\|_{TV} = \|\sum_{k=1}^{B} (P^{\otimes(B-k+1)} \otimes Q^{\otimes(k-1)} - P^{\otimes(B-k)} \otimes Q^{\otimes k})\|_{TV}$$

$$\leq \sum_{k=1}^{B} \|P^{\otimes(B-k+1)} \otimes Q^{\otimes(k-1)} - P^{\otimes(B-k)} \otimes Q^{\otimes k}\|_{TV} \tag{107}$$

$$\leq \sum_{k=1}^{B} \|P^{\otimes(B-k)} \otimes P \otimes Q^{\otimes(k-1)} - P^{\otimes(B-k)} \otimes Q \otimes Q^{\otimes(k-1)}\|_{TV}$$

$$\leq \sum_{k=1}^{B} \|P - Q\|_{TV} = B\|P - Q\|_{TV}, \tag{108}$$

where (107) follows from the triangle inequality for the TVD ; and (108) follows form Lemma H.2, with $W = X_{B-k+1}$, $\Pi_W \equiv P$ and $\Gamma_W \equiv Q$.

## I   THE BIRTHDAY PARADOX

We provide a proof of Claim B.2. We have

$$
\begin{aligned}
(p^{\mathcal{U}}_{[\lfloor 2^{R_1}\rfloor]})^{\otimes B}\big(M^{(1)}, ..., M^{(B)} \text{ 2 by 2 distinct}\big) &= \prod_{k=1}^{B} \frac{\lfloor 2^{R_1}\rfloor - k + 1}{\lfloor 2^{R_1}\rfloor} \\
&\geq \frac{(\lfloor 2^{R_1}\rfloor - B + 1)^B}{\lfloor 2^{R_1}\rfloor^B} \\
&\geq \Big(1 - \frac{B-1}{\lfloor 2^{R_1}\rfloor}\Big)^B \\
&\geq 1 - \frac{B(B-1)}{\lfloor 2^{R_1}\rfloor} \\
&\geq 1 - \frac{B^2}{\lfloor 2^{R_1}\rfloor},
\end{aligned}
\tag{109}
$$

where (109) follows from Bernoulli's inequality, since $R_1 > \log(B)$.

## J   EXISTENCE OF A UNIVERSAL $p^{\otimes *}$-CRITIC

We provide a proof of Theorem A.4. From Li & Vitányi (2019, Theorem 4.3.1), there exists a sequence $\{q_n\}_{n \geq 1}$ containing all lower semi-computable semi-measures on $\{0,1\}^*$, and a sequence $\{\pi_n\}_{n \geq 1}$ of (strictly) positive reals, such that the mixture defined by

$$
\mathbf{m} := \sum_{n \geq 1} \pi_n q_n
\tag{110}
$$

is a lower semi-computable semi-measure on $\{0,1\}^*$. For every $n \in \mathbb{N}$, let $\mathbf{m}(\mathcal{X}^n)$ denote

$$
\sum_{x_{1:n} \in \mathcal{X}^n} \mathbf{m}(x_{1:n}).
$$

From (110), we have $\forall x \in \{0,1\}^*, \mathbf{m}(x) > 0$. Moreover, $\forall x_0 \in \mathcal{X}, \ p(x_0) > 0$, thus $\forall x \in \cup_{n \in \mathbb{N}} \mathcal{X}^n, \ p^{\otimes *}(x) > 0$. Define function $\delta_0$, by

$$
\forall n \in \mathbb{N}, \forall x_{1:n} \in \mathcal{X}^n, \ \delta_0(x_{1:n}) := \log\Big(\frac{\mathbf{m}(x_{1:n})}{\mathbf{m}(\mathcal{X}^n)p^{\otimes n}(x_{1:n})}\Big).
\tag{111}
$$

Fix any lower semi-computable $p^{\otimes *}$-critic $\delta$. Define map $q_\delta : \{0,1\}^* \to \mathbb{R}$ by

$$
\forall x \in \cup_{n \in \mathbb{N}} \mathcal{X}^n, \quad q_\delta(x) := \mathbf{m}(\mathcal{X}^{l(x)}) 2^{\delta(x)} p^{\otimes *}(x),
$$

and $x \mapsto 0$ elsewhere. From Lemma A.2 (iii), the function which is null outside of $\cup_{n \in \mathbb{N}} \mathcal{X}^n$, and defined by $x \mapsto \mathbf{m}(\mathcal{X}^{l(x)})$ on $\cup_{n \in \mathbb{N}} \mathcal{X}^n$, is lower semi-computable. Moreover, $x \mapsto 2^{\delta(x)}$ and $x \mapsto p^{\otimes *}(x)$ are lower semi-computable by Lemma A.2 (i) and (iii) respectively. Thus, $q_\delta$ is the product three non-negative lower semi-computable functions. Hence, $q_\delta$ is lower semi-computable by Lemma A.2 (i). Moreover, we have

$$
\begin{aligned}
\sum_{x \in \{0,1\}^*} q_\delta(x) &= \sum_{n \in \mathbb{N}} \mathbf{m}(\mathcal{X}^n) \sum_{x \in \mathcal{X}^n} 2^{\delta(x)} p^{\otimes n}(x) \\
&\leq \sum_{n \in \mathbb{N}} \mathbf{m}(\mathcal{X}^n) \\
&\leq 1,
\end{aligned}
\tag{112}
\tag{113}
$$

where (112) follows from the definition of a $p^{\otimes *}$-critic; and (113) follows from the fact that $\mathbf{m}$ is a semi-measure. Therefore, $q_\delta$ is a lower semi-computable semi-measure. Thus, from (110), we have $\mathbf{m} \geq \pi_{q_\delta} q_\delta$, for some positive real $\pi_{q_\delta}$. In order to derive (23), fix $x \in \cup_{n \in \mathbb{N}} \mathcal{X}^n$, and denote $l(x)$ by

$n$. From (110), we have $\mathbf{m}(x) > 0$. Therefore, since $\forall x_0 \in \mathcal{X}, p(x_0) > 0$, we have $q_\delta(x) > 0$. Thus, from (111), we have

$$\delta_0(x) = \log\left(\frac{\mathbf{m}(x_{1:n})}{\mathbf{m}(\mathcal{X}^n)p^{\otimes n}(x_{1:n})}\right)$$
$$\geq \log\left(\frac{\pi_{q_\delta}q(x_{1:n})}{\mathbf{m}(\mathcal{X}^n)p^{\otimes n}(x_{1:n})}\right)$$
$$= \log(\pi_{q_\delta}) + \delta(x).$$

This is true for any lower semi-computable $p^{\otimes *}$-critic $\delta$, and any $x \in \cup_{n\in\mathbb{N}}\mathcal{X}^n$. Since $\log(\pi_{q_\delta})$ does not depend on $x$, then property (23) holds. This concludes the proof.

## K  ADDITIONAL SEMI-COMPUTABILITY ARGUMENTS

We provide a proof of Lemma A.2. If $f$ is lower semi-computable, we denote by $(x, k) \mapsto \varphi_{f,-}(x, k)$ a computable function from $\mathcal{E}$ to $\mathbb{Q}$, monotonically approaching $f$ from below, in the sense of Definition A.1. If $f$ is upper semi-computable, then $\varphi_{f,+}(x, k)$ denotes a function of the form $\varphi_{-f,-}(x, k)$, which monotonically approaches $f$ from above.

### K.1  ASSUME THAT $f$ AND $g$ ARE COMPUTABLE

#### K.1.1  $f + g$

Function $\varphi_{f,-} + \varphi_{g,-}$ is a computable function from $\mathcal{E} \times \mathbb{N}$ to $\mathbb{Q}$, which monotonically approaches $f + g$ from below. Similarly, $\varphi_{f,+} + \varphi_{g,+}$ constitutes a computable rational upper approximation.

#### K.1.2  $|f|$

We construct $\varphi_{|f|,-}$ as follows. Let $x \in \mathcal{E}$ and $k \in \mathbb{N}$. If $\varphi_{f,-}(x, k) \geq 0$, return $|\varphi_{f,-}(x, k)|$. Otherwise, if $\varphi_{f,+}(x, k) \leq 0$, return $|\varphi_{f,+}(x, k)|$. Otherwise, return $0$. We define $\varphi_{|f|,+}(x, k)$ as

$$\max\left(|\varphi_{f,-}(x, k)|, |\varphi_{f,+}(x, k)|\right).$$

Straightforwardly, this implies that $|f|$ is computable.

#### K.1.3  $fg$

Define $\varphi_{fg,-}(x, k)$ as follows. If $\varphi_{f,-}(x, k) \geq 0$ and $\varphi_{g,-}(x, k) \geq 0$, then return $\varphi_{f,-}(x, k)\varphi_{g,-}(x, k)$. Otherwise, if $\varphi_{f,+}(x, k) \leq 0$ and $\varphi_{g,+}(x, k) \leq 0$, then return $\varphi_{f,+}(x, k)\varphi_{g,+}(x, k)$. Otherwise, return

$$-\max\left(|\varphi_{f,-}(x, k)|, |\varphi_{f,+}(x, k)|\right)\max\left(|\varphi_{g,-}(x, k)|, |\varphi_{g,+}(x, k)|\right).$$

Define $\varphi_{fg,+}$ as $-\varphi_{(-f)g,-}$.

### K.2  SUPPOSE THAT $f$ IS COMPUTABLE AND ONLY TAKES POSITIVE VALUES

#### K.2.1  $1/f$

Define $\varphi_{1/f,-}(x, k)$ as $1/\varphi_{f,+}(x, k)$. Compute $k_1(x)$, the smallest positive integer $k$ such that $\varphi_{f,-}(x, k) > 0$. For all integers $k \in [1, k_1(x)]$, define $\varphi_{1/f,+}(x, k)$ as $1/\varphi_{f,-}(x, k_1(x))$. For all integers $k \in (k_1(x), \infty)$, define $\varphi_{1/f,+}(x, k)$ as $1/\varphi_{f,-}(x, k)$.

#### K.2.2  $f^{1/b}$

Compute $k_1(x)$, the smallest positive integer $k$ such that $\varphi_{f,-}(x, k) > 0$. For all integers $k \in [1, k_1(x))$, define $\varphi_{f^{1/b},-}(x, k) := 0$ and $\varphi_{f^{1/b},+}(x, k) = \lceil\varphi_{f,+}(x, k)\rceil$. Consider an integer $k \in [k_1(x), \infty)$. Compute the greatest integer $m$ such that $(m/2^k)^b \leq \varphi_{f,-}(x, k)$. Then, define

$\varphi_{f^{1/b},-}(x,k) := m/2^k$. Therefore, we have

$$\forall k \geq k_1(x), \ \ 0 \leq \varphi_{f,-}(x,k)^{1/b} - \varphi_{f^{1/b},-}(x,k) < \frac{1}{2^k}. \tag{114}$$

From (114), and since the $b$-th root function and $k \mapsto \varphi_{f,-}(x,k)$ are both non-decreasing, we have

$$\forall k \geq k_1(x) + 1, \ \ \varphi_{f^{1/b},-}(x,k-1) \leq \varphi_{f,-}(x,k)^{1/b}. \tag{115}$$

Since $\varphi_{f^{1/b},-}(x,k-1)$ can also be written in the form $m'/2^k$, then, from the maximality of the integer $m$ appearing in the construction of $\varphi_{f^{1/b},-}(x,k)$, we have

$$\forall k \geq k_1(x) + 1, \ \ \varphi_{f^{1/b},-}(x,k-1) \leq \varphi_{f^{1/b},-}(x,k). \tag{116}$$

This also holds for all integers $k \in [2, k_1(x)+1]$. Properties (114) and (116) imply that $f^{1/b}$ is lower semi-computable. We prove upper semi-computability similarly, using the smallest integer $\tilde{m}$ such that $(\tilde{m}/2^k)^b \geq \varphi_{f,+}(x,k)$, and setting $\varphi_{f^{1/b},+}(x,k) := \tilde{m}/2^k$.

### K.3 Assume that $f$ and $g$ are lower semi-computable

#### K.3.1 $f + g$

Function $\varphi_{f,-} + \varphi_{g,-}$ is a computable function from $\mathcal{E} \times \mathbb{N}$ to $\mathbb{Q}$, which monotonically approaches $f + g$ from below.

#### K.3.2 $\lceil f \rceil$

Define $\varphi_{\lceil f \rceil,-}$ as $\lceil \varphi_{f,-} \rceil$.

#### K.3.3 $2^f$

Fix $x \in \mathcal{E}$ and $k \in \mathbb{N}$. Let $a \in \mathbb{Z}$ and $b \in \mathbb{N}$ such that $\varphi_{f,-}(x,k) = a/b$. Compute the greatest integer $m$ such that $(m/2^k)^b \leq 2^a$. Then, define $\varphi_{2^f,-}(x,k) := m/2^k$. Therefore, we have

$$0 \leq 2^{\varphi_{f,-}(x,k)} - \varphi_{2^f,-}(x,k) < \frac{1}{2^k}. \tag{117}$$

From (117), and since the exponential function and $k \mapsto \varphi_{f,-}(x,k)$ are both non-decreasing, we have

$$\forall k \geq 2, \ \varphi_{2^f,-}(x,k-1) \leq 2^{\varphi_{f,-}(x,k)}. \tag{118}$$

Since $\varphi_{2^f,-}(x,k-1)$ can also be written in the form $m'/2^k$, then, from the maximality of the integer $m$ appearing in the construction of $\varphi_{2^f,-}(x,k)$, we have

$$\forall k \geq 2, \ \varphi_{2^f,-}(x,k-1) \leq \varphi_{2^f,-}(x,k). \tag{119}$$

Properties (117) and (119) imply that $2^f$ is lower semi-computable.

### K.4 Assume that $f$ and $g$ are semi-computable and non-negative

#### K.4.1 $fg$

If $\varphi_{f,-}(x,k) \geq 0$ and $\varphi_{g,-}(x,k) \geq 0$, return $\varphi_{f,-}(x,k)\varphi_{g,-}(x,k)$. Otherwise, return 0.

#### K.4.2 $2^f/(3+f)^2$

There exists a real $\varepsilon \in (0,1)$ such that $u \mapsto 2^u/(3+u)^2$ is non-decreasing on $(-\varepsilon, \infty)$. Fix $x \in \mathcal{E}$. Compute $k_1(x)$, the smallest positive integer $k$ such that $\varphi_{f,-}(x,k) > -\varepsilon$. For all integers $k \in [1, k_1(x))$, define $\varphi_{2^f/(3+f)^2,-}(x,k) := 0$. Fix an integer $k \geq k_1(x)$. Let $a \in \mathbb{Z}$ and $b \in \mathbb{N}$ such that $\varphi_{f,-}(x,k) = a/b$. Compute the greatest integer $m$ such that $(m/2^k)^b \leq 2^a/(3+a/b)^{2b}$. Then, define $\varphi_{2^f/(3+f)^2,-}(x,k) := m/2^k$. Therefore, we have

$$\forall k \geq k_1(x), \ \ 0 \leq \frac{2^{\varphi_{f,-}(x,k)}}{(3+\varphi_{f,-}(x,k))^2} - \varphi_{2^f/(3+f)^2,-}(x,k) < \frac{1}{2^k}. \tag{120}$$

From (120), and since $k \mapsto \varphi_{f,-}(x, k)$ is non-decreasing, and $u \mapsto 2^u/(3+u)^2$ is non-decreasing on $(-\varepsilon, \infty)$, we have

$$\forall k \geq k_1(x) + 1, \quad \varphi_{2^f/(3+f)^2,-}(x, k-1) \leq \frac{2^{\varphi_{f,-}(x,k)}}{(3 + \varphi_{f,-}(x,k))^2}. \tag{121}$$

Since $\varphi_{2^f/(3+f)^2,-}(x, k-1)$ can also be written in the form $m'/2^k$, then, from the maximality of the integer $m$ appearing in the construction of $\varphi_{2^f/(3+f)^2,-}(x, k)$, we have

$$\forall k \geq k_1(x) + 1, \quad \varphi_{2^f/(3+f)^2,-}(x, k-1) \leq \varphi_{2^f/(3+f)^2,-}(x, k). \tag{122}$$

This is also true for all integers $k \in [2, k_1(x) + 1]$. Properties (120) and (122) imply that $2^f/(3+f)^2$ is lower semi-computable.

### K.4.3 $\log(f)$

Assume that $f$ only takes positive values. Fix $x \in \mathcal{E}$. Compute $k_1(x)$, the smallest positive integer $k$ such that $\varphi_{f,-}(x, k) > 0$. Fix an integer $k \geq k_1(x)$. Compute the largest integer $m$ such that $2^m \leq \varphi_{f,-}(x, k)^{2^k}$. Then, define $\varphi_{\log(f),-}(x, k) := m/2^k$. For all integers $k \in [1, k_1(x))$, define $\varphi_{\log(f),-}(x, k)$ as $\varphi_{\log(f),-}(x, k_1(x))$. Therefore, we have

$$\forall k \geq k_1(x), \quad 0 \leq \log(\varphi_{f,-}(x, k)) - \varphi_{\log(f),-}(x, k) < \frac{1}{2^k}. \tag{123}$$

From (123), and since the logarithm and $k \mapsto \varphi_{f,-}(x, k)$ are both non-decreasing, we have

$$\forall k \geq k_1(x) + 1, \quad \varphi_{\log(f),-}(x, k-1) \leq \log(\varphi_{f,-}(x, k)). \tag{124}$$

Since $\varphi_{\log(f),-}(x, k-1)$ can also be written in the form $m'/2^k$, then, from the maximality of the integer $m$ appearing in the construction of $\varphi_{\log(f),-}(x, k)$, we have

$$\forall k \geq k_1(x) + 1, \quad \varphi_{\log(f),-}(x, k-1) \leq \varphi_{\log(f),-}(x, k). \tag{125}$$

This also holds for all integers $k \in [2, k_1(x) + 1)$. Properties (123) and (125) imply that $\log(f)$ is lower semi-computable.

### K.5 FUNCTIONS OF FINITE BINARY STRINGS

Let $\mathcal{X}$ be a finite computable subset of $\{0, 1\}^*$, and $f$ be a lower semi-computable function from $\{0, 1\}^*$ into $\mathbb{R}$.

**Lemma K.1.** $\cup_{n \in \mathbb{N}} \mathcal{X}^n$ *is a computable set.*

*Proof.* By Definition A.1, it is sufficient to construct a computable function $\tau$ from $\{0, 1\}^*$ to $\{0, 1\}$, which returns 1 if its input is in $\cup_{n \in \mathbb{N}} \mathcal{X}^n$, and 0 otherwise. Since $\mathcal{X}$ is computable, there exists a computable function $\tau_0$ from $\{0, 1\}^*$ to $\{0, 1\}$, which returns 1 if its input is in $\mathcal{X}$, and 0 otherwise. Fix $x \in \{0, 1\}^*$. Define $\tau(x)$ as follows. Enumerate all partitions of $x$ into consecutive sub-strings. For each, call $\tau_0$ on every sub-string. If for some partition, the output of $\tau_0$ is 1 for every sub-string, then return 1. Otherwise, return 0. $\square$

Hereafter, we use the notation $\tau$ defined in the above proof.

#### K.5.1 PARTIAL SUMS

Consider the function $\tilde{f} : \{0, 1\}^* \to \mathbb{R}$ which is null outside of $\cup_{n \in \mathbb{N}} \mathcal{X}^n$, and is defined by

$$\forall x \in \cup_{n \in \mathbb{N}} \mathcal{X}^n, \quad \tilde{f}(x) = \sum_{y \in \mathcal{X}^{l(x)}} f(y).$$

Fix $x \in \{0, 1\}^*$ and $k \in \mathbb{N}$. Define $\varphi_{\tilde{f},-}(x, k)$ as follows. Compute $\tau(x)$. If it is null, return 0. Otherwise: compute $l(x)$, and for each $y$ in $\mathcal{X}^{l(x)}$, compute $\varphi_{f,-}(y, k)$, then return

$$\sum_{y \in \mathcal{X}^{l(x)}} \varphi_{f,-}(y, k).$$

Fix some $x \in \cup_{n \in \mathbb{N}} \mathcal{X}^n$. The (finite) set of indices of the above sum does not depend on $k$. Therefore, since for each $y \in \{0, 1\}^*$ we have $\varphi_{f,-}(y, k) \underset{k \to \infty}{\to} f(y)$, we get

$$\forall x \in \cup_{n \in \mathbb{N}} \mathcal{X}^n, \ \varphi_{\tilde{f},-}(x, k) \underset{k \to \infty}{\longrightarrow} \tilde{f}(x). \tag{126}$$

Similarly, since for any $y \in \{0, 1\}^*$ and any $k \geq 1$, we have $\varphi_{f,-}(y, k) \leq \varphi_{f,-}(y, k+1)$, then we have

$$\forall x \in \cup_{n \in \mathbb{N}} \mathcal{X}^n, \forall k \in \mathbb{N}, \ \varphi_{\tilde{f},-}(x, k) \leq \varphi_{\tilde{f},-}(x, k+1). \tag{127}$$

Properties (126) and (127) also hold for finite strings outside of $\cup_{n \in \mathbb{N}} \mathcal{X}^n$. Thus, $\tilde{f}$ is lower semi-computable.

### K.5.2 PRODUCT DISTRIBUTION

Let $p$ be a lower semi-computable probability measure on $\mathcal{X}$. Fix $x \in \{0, 1\}^*$ and $k \in \mathbb{N}$. Define $\varphi_{p^{\otimes*},-}(x, k)$ as follows. Compute $\tau(x)$. If it is null, return 0. Otherwise, proceed as follows. Compute $l(x)$. We write $x$ as $x_{1:l(x)}$, with $x_t \in \mathcal{X}$ for any integer $t$ in $[1, l(x)]$. Compute and return

$$\prod_{t=1}^{l(x)} \varphi_{p,-}(x_t, k).$$

Fix some $x \in \cup_{n \in \mathbb{N}} \mathcal{X}^n$. The (finite) set of indices of the above product does not depend on $k$. Therefore, since for each $y \in \mathcal{X}$, we have $\varphi_{p,-}(y, k) \underset{k \to \infty}{\to} p(y)$, we get

$$\forall x \in \cup_{n \in \mathbb{N}} \mathcal{X}^n, \ \varphi_{p^{\otimes*},-}(x, k) \underset{k \to \infty}{\longrightarrow} p^{\otimes*}(x). \tag{128}$$

Similarly, since for any $y \in \mathcal{X}$, and any $k \geq 1$, we have $\varphi_{p,-}(y, k) \leq \varphi_{p,-}(y, k+1)$, and since $p$ is non-negative, then we have

$$\forall x \in \cup_{n \in \mathbb{N}} \mathcal{X}^n, \forall k \in \mathbb{N}, \ \varphi_{p^{\otimes*},-}(x, k) \leq \varphi_{p^{\otimes*},-}(x, k+1). \tag{129}$$

Properties (128) and (129) also hold for finite strings outside of $\cup_{n \in \mathbb{N}} \mathcal{X}^n$. Thus, $p^{\otimes*}$ is lower semi-computable.

