# OpenReview forum: "The Rate-Distortion-Perception Trade-Off with Algorithmic Realism"
_ICLR.cc/2025/Conference — Submitted to ICLR 2025_

### Official Review · Reviewer_Axnd · 2024-10-21

**Soundness:** 2
**Presentation:** 3
**Contribution:** 4
**Rating:** 6
**Confidence:** 4

**Summary:**

This paper proposes a new mathematical formulation for the rate-perception-distortion tradeoff. Specifically, in the previous rate-perception-distortion formulation, the perceptual quality constraint is a constraint on the statistical divergence between the distribution of the decoded images and that of the clean images. In theory, this typically leads to randomized decoders, which produce many different decoded images given an encoded one. However, in practice, high-perceptual-quality compression-decompression algorithms rarely incorporate such randomness.
To explain this phenomenon, the authors replace the perceptual quality constraint with a new interesting concept called the "universal critic", which poses a perceptual quality constraint on individual images (or on a batch of images).
The new rate-perception-distortion formulation leads to solutions which do not incorporate randomness. This is a sensible result given the fact that now there is no constraint on the *distribution* of the decoded images.

**Strengths:**

This paper is incredibly interesting, and written very well. The theoretical results are interesting and serve a highly important contribution to the community of information theorists.

**Weaknesses:**

1. There are no experiments, demonstrations, simulations, presented evidence, etc. This paper contains only theoretical results, which is not necessarily a bad thing, but I am not sure whether it's a fit for the ICLR community (most of which are practitioners). I would expect to see this paper in a theoretical journal.

2. There is no discussion/limitation section discussing the possible future continuation of this work.

**Questions:**

1. I am aware that universal critics cannot be implemented practically, but still, is there a way to somehow simulate/demonstrate the new tradeoff on simple examples (perhaps a Bernoulli source)?

2. Is there a way to demonstrate on previous works that less randomness is indeed attributed to better universal critic scores? Namely, is it possible to demonstrate that one may benefit from better rate&distortion by avoiding randomness, but still achieving high-perceptual-quality in the sense of universal critics?

---

> ### Author Response · Authors · 2024-11-25
>
> Response to the “Weaknesses” section of the review:
>
> 1. We consider that the theoretical results in the paper, regarding the role of common randomness, are supported by the existing literature on practical
> generative compression algorithms. Indeed, there exist compression schemes
> (Agustsson et al.; He et al., 2022; Hoogeboom et al., 2023; Ghouse et al., 2023;
> Mentzer et al., 2020; Yang & Mandt, 2023), considered as state-of-the-art in terms of the trade-off between rate, distortion, and realism, that do not involve any common randomness.
>
> 2. The following is a possible avenue for future research. Our work theoretically shows that the number of samples
> inspected by a critic greatly influences the trade-off between rate, distortion, and
> realism. This constitutes a new perspective on the latter. Therefore, our work
> may inspire investigations of metrics used to quantify realism, where particular
> attention would be given to the choice of the batch size. This could lead to
> highlighting specific strengths and weaknesses of existing realism metrics. It
> may also inspire a critical assessment of the relative performance of existing
> compression schemes, depending on the choice of realism metric.
>
> Response to the “Questions” section of the review:
>
> 1. As the reviewer pointed out, our formulation involves a very strong critic, stronger
> than can be implemented in practice. This is needed to conclusively show that
> randomness is not needed to fool critics, which is our main result. Therefore,
> albeit being tailored to this goal, it is not directly tailored for practical
> implementation purposes. After careful consideration, we have reached the
> conclusion that the core concept of our paper, namely the optimality of
> deterministic codes, cannot be illustrated by examples. Indeed, we are not
> aware of any experimental method allowing to ascertain the optimality of a given
> compression algorithm (at a given target rate) for non-trivial choices of source
> and distortion measure. Moreover, our characterization of this new trade-off in
> the asymptotic setting applies to a Bernoulli source. However, we are not able to
> derive further results for such a source, since characterizing the trade-off in a
> non-asymptotic setting is a challenging problem.
>
> 2. We thank the reviewer for this suggestion. However, the question of whether
> randomization has an adverse effect in terms of rate, or distortion, or universal
> critic scores, remains an open problem. In this paper, we have only shown that
> there exist optimal deterministic schemes.

---

> > ### Comment · Reviewer_Axnd · 2024-11-29
> >
> > I thank the authors for their response.
> > I believe this paper is valuable. But given the fact that it is not practical, cannot be implemented, cannot properly be demonstrated and verified, illustrated, etc, I cannot confidently recommend acceptance.
> > I will therefore keep my score.

---

> > > ### Author Response · Authors · 2024-12-04
> > >
> > > We would like to clarify once again that, we did not choose not to include simulation results simply because we are not able to run experiments. It is because running numerical experiments would not actually add any value to the claim of this paper.
> > >
> > > Maybe we can explain this with an analogy to security research. In security, to claim that a scheme is provably secure, it is not sufficient to show that a particular attack fails. One can simulate as many attacks as possible, it would still not be possible to make any theoretical claims. Rather, it is necessary to prove that ANY attack will fail, however high the latter’s computational complexity is. Therefore, proving that a scheme is secure is done by considering the strongest possible attacks, with unlimited computational complexity, and showing theoretically that all such attacks would fail.
> > >
> > > Similarly, in our work, to prove that common randomness is not needed to fool critics, it is not sufficient to establish this for a particular critic that has been experimentally validated, since the conclusion would not extend to other critics that might be proposed later. One must consider a very strong critic, stronger than any that can be implemented in practice. Therefore, including such simulations in our paper would not add anything to our claim.
> > >
> > > Given that this is an application area that is of significant interest to the ML community, we believe this theoretical result would be of value to researchers in this community. It provides a fresh perspective to many results in the recent literature that deal with realism from an information theoretic perspective, and answers (partially) the question regarding the role common randomness plays in these results.
> > >
> > > Moreover, we would like to bring to the attention of the reviewer that many high-quality theoretical papers without any experiments or toy theoretical examples have appeared in machine learning conferences. The following are a few examples, among many:
> > >
> > > Unified Lower Bounds for Interactive High-dimensional Estimation under Information Constraints, NeurIPS 2023
> > >
> > > Distributed Learning with Sublinear Communication, ICML 2021
> > >
> > > Learning Policies with Zero or Bounded Constraint Violation for Constrained MDPs, NeurIPS 2021
> > >
> > > Optimal Algorithms for Non-Smooth Distributed Optimization in Networks, NeurIPS 2018

---

### Official Review · Reviewer_EaDh · 2024-10-30

**Soundness:** 3
**Presentation:** 3
**Contribution:** 3
**Rating:** 6
**Confidence:** 4

**Summary:**

This paper propose a new rate-distortion-perception function and proves its achievabiliy and converse, in both zero-shot and asymptotical setting. The proposed RDP function replace the P from divergence to a realism measure defined by authors. The propose RDP function is achievable without common randomness.

**Strengths:**

It is great to have a RDP which is achievable without randomness. Afterall, the human eye distinguishs images in a per-image setting without randomness. The proposed RDP is better aligned to human perception in this sense. I have not went through the details of proofs due to the complex notation. However, I am in general glad to see a new RDP function with achievability & converse, zero-shot & asymptotic.

**Weaknesses:**

The reason why I am not willing to give this paper a higher rating is that the authors have not shown how the proposed RDP can guide perceptual compression / super-resolution, not even a toy example.

The RDP function in [Blau & Michaeli 2019] has many disadvantages, which this paper does not have:
* [Blau & Michaeli 2019] does not prove the converse.
* [Blau & Michaeli 2019] does not distinguish zero-shot and asymptotic function.
Those issues have not been fixed until [A coding theorem for the rate-distortion-perception function].

However, those weakness does not stop [Blau & Michaeli 2019] being popular. This is because [Blau & Michaeli 2019] has clear application in perceptual compression / super-resolution. It explains why previous work using GAN for perceptual compression; It aligns very well with the practically used "real vs. fake" test; It even guides later works in diffusion based image compression.

ICLR is a machine learning venue, not a pure information theory venue such as ISIT / TIT. It is better to have numerical examples (even toy size) and suggestions for later application works, so that the later works in ICLR can benefits from this paper more.

(minor) It is better to move the converse to the main paper, as this year we have 10 page budget. It is strange to have a 8 page paper and 20 page appendix. At least for me, the converse is as important as achievability.

**Questions:**

See weakness.

---

> ### Author Response · Authors · 2024-11-25
>
> Our work theoretically shows that the number of samples inspected by a critic greatly
> influences the trade-off between rate, distortion, and realism. This constitutes a new
> perspective on the latter. Therefore, our work may inspire investigations of metrics used
> to quantify realism, where particular attention would be given to the choice of the batch
> size. This could lead to highlighting specific strengths and weaknesses of existing
> realism metrics. It may also inspire a critical assessment of the relative performance of
> existing compression schemes, depending on the choice of realism metric.

---

> > ### Comment · Reviewer_EaDh · 2024-11-26
> >
> > Thanks for the rebuttal. I keep my initial rating and support accepting this paper.

---

### Official Review · Reviewer_hK1x · 2024-11-01

**Soundness:** 3
**Presentation:** 2
**Contribution:** 2
**Rating:** 5
**Confidence:** 2

**Summary:**

The study addresses a core issue in lossy image compression: achieving high perceptual quality in the decompressed images while minimizing distortion and compression rate. A unique aspect of this paper is its focus on algorithmic realism — a concept that considers human perception and aims to create compressed images that appear realistic to a critic. This builds on prior work on the rate-distortion-perception (RDP) trade-off, but instead of relying heavily on common randomness, it introduces a framework that reduces or eliminates the need for shared randomness between encoder and decoder in practical settings.

**Strengths:**

1. Interesting Insight into Realism Constraints: By redefining perceptual realism through an algorithmic lens, the paper provides a fresh perspective on the RDP trade-off and its practical applications in lossy compression.
2. Reduced Dependency on Common Randomness: The finding that common randomness is only needed in impractically large batches addresses a significant gap in previous theoretical predictions versus experimental observations.
3. Good Theoretical Foundation: The study provides rigorous proof and aligns well with information theory, making it a valuable resource for researchers interested in theoretical advances in compression.

**Weaknesses:**

While the paper provides rigorous theoretical derivations and proofs, one significant limitation is the lack of practical illustrations or implementations that could help readers appreciate the impact and contributions of the proposed framework in real-world applications. The authors claim that algorithmic realism simplifies the practical attainment of the rate-distortion-perception (RDP) trade-off by reducing the dependency on common randomness between encoder and decoder. However, without practical visualizations or demonstration attempts, it becomes challenging for readers to intuitively evaluate the work’s contributions.

Though I acknowledge the value of theoretical derivations, the paper appears incomplete and, consequently, less persuasive without empirical validation. I strongly recommend that the authors complement their theoretical results with practical experiments, such as specific implementations, visual examples, or a demonstration. This would significantly enhance the paper’s credibility and provide readers with a tangible understanding of the theory’s implications.

To make these points more specific, I propose the following questions:

 **1. Evaluation of Practical Applicability**

The paper offers extensive theoretical proofs, yet there is no concrete implementation provided to demonstrate how this framework could be integrated into real-world image compression tasks. Could the authors consider validating the proposed approach on an actual compression system to illustrate its practical efficacy?

 **2. Feasibility of Reducing Common Randomness.**

While the theory is sound, it would benefit from an empirical investigation to verify that reducing common randomness does not detract from visual quality. Without experimental validation, how can readers assess the applicability of these theoretical findings to practical compression systems?

   **3. Experimental Support for Theory-Practice Connection**

 The paper’s theoretical framework is detailed but lacks experimental applications or use cases. Could the authors consider providing experiments to demonstrate the balance of visual quality and compression rate achieved by the proposed approach?

   **4. Inclusion of Visual Case Studies**

 Given the claims of practical feasibility, is it possible to provide specific examples of compressed and decompressed images to offer readers a more direct perception of the quality improvement achieved by the proposed approach?

These additions would substantially enhance the paper by bridging the gap between theoretical results and their practical impact, allowing readers to more fully appreciate the contributions of this work.

**Questions:**

Please see the weakness.

---

> ### Author Response · Authors · 2024-11-25
>
> For the sake of clarity, and since we have identified links between the reviewer’s
> questions, we will answer the latter in a different order.
>
> Response to questions 2, 3 and 4: These questions pertain to the impact, on
> visual/perceptual quality and compression rate, of designing compression schemes
> which do not involve common randomness. We have proved theoretically that such
> schemes can achieve the optimal trade-off between realism, rate, and distortion. This is
> supported by the vast literature on practical generative compression algorithms.
> Indeed, there exist compression schemes (Agustsson et al.; He et al., 2022;
> Hoogeboom et al., 2023; Ghouse et al., 2023; Mentzer et al., 2020; Yang & Mandt, 2023),
> considered as state-of-the-art in terms of the trade-off between rate, distortion, and realism, that do not involve any common randomness.
>
> Response to question 1: Our claim is that deterministic schemes can achieve the
> optimal trade-off between rate, distortion, and realism. To conclusively show that
> randomness is not needed to fool a critic, it was necessary to consider the strongest
> critic possible, stronger than can be implemented in practice. Therefore, this
> framework, albeit being tailored to our theoretical proof, is not directly tailored for
> integration into a practical compression scheme. The design of practical perceptual
> quality critics, with adequate batch sizes, is thus out of the scope of this paper, but
> constitutes an interesting avenue for future work.

---

> > ### Comment · Reviewer_hK1x · 2024-11-26
> > **Reponse to Authors**
> >
> > Thanks for the response. I am unsure whether it is acceptable to exclude any experiments entirely and focus solely on the theoretical aspects for an ICLR submission.
> >
> > While theoretical contributions are important, readers (especially from machine learning society) typically expect to see experimental results that validate the effectiveness and practical applicability of the theory.
> > Providing only the theory without any experimental validation may lead readers to feel that the work lacks sufficient support or empirical evidence.
> >
> > Therefore, I would like to maintain my rating and lower my confidence.

---

> > > ### Author Response · Authors · 2024-12-04
> > >
> > > We would like to clarify once again that, we did not choose not to include simulation results simply because we are not able to run experiments. It is because running numerical experiments would not actually add any value to the claim of this paper.
> > >
> > > Maybe we can explain this with an analogy to security research. In security, to claim that a scheme is provably secure, it is not sufficient to show that a particular attack fails. One can simulate as many attacks as possible, it would still not be possible to make any theoretical claims. Rather, it is necessary to prove that ANY attack will fail, however high the latter’s computational complexity is. Therefore, proving that a scheme is secure is done by considering the strongest possible attacks, with unlimited computational complexity, and showing theoretically that all such attacks would fail.
> > >
> > > Similarly, in our work, to prove that common randomness is not needed to fool critics, it is not sufficient to establish this for a particular critic that has been experimentally validated, since the conclusion would not extend to other critics that might be proposed later. One must consider a very strong critic, stronger than any that can be implemented in practice. Therefore, including such simulations in our paper would not add anything to our claim.
> > >
> > > Given that this is an application area that is of significant interest to the ML community, we believe this theoretical result would be of value to researchers in this community. It provides a fresh perspective to many results in the recent literature that deal with realism from an information theoretic perspective, and answers (partially) the question regarding the role common randomness plays in these results.
> > >
> > > Moreover, we would like to bring to the attention of the reviewer that many high-quality theoretical papers without any experiments or toy theoretical examples have appeared in machine learning conferences. The following are a few examples, among many:
> > >
> > > Unified Lower Bounds for Interactive High-dimensional Estimation under Information Constraints, NeurIPS 2023
> > >
> > > Distributed Learning with Sublinear Communication, ICML 2021
> > >
> > > Learning Policies with Zero or Bounded Constraint Violation for Constrained MDPs, NeurIPS 2021
> > >
> > > Optimal Algorithms for Non-Smooth Distributed Optimization in Networks, NeurIPS 2018

---

### Official Review · Reviewer_omKD · 2024-11-03

**Soundness:** 3
**Presentation:** 2
**Contribution:** 2
**Rating:** 5
**Confidence:** 4

**Summary:**

The paper concerns with the rate-distortion-perception tradeoff (RDP) in the context of lossy compression and argues that previous theoretical results, which suggest that common randomness between the encoder and the decoder is crucial for good performance, do not accurately reflect how humans perceive realism. To address this, the authors reformulate the RDP with reaslim constraints by adopting the concept of universal critic that generalizes no-reference metrics and divergences and insecpt batches of samples. Under this framework, they prove that near-perfect realism is achievable without common randomness unless the batch size is impractically large and the proposed realism measure reduces to a divergence.

**Strengths:**

* The paper provides a novel perspective on the rate-distortion-perception tradeoff by adopting the concept of universal critics.
* The paper presents rigorous theoretical analysis and proofs to support its claims.
* The theoretical finding that near-perfect realism is achievable without common randomness has significant practical implications for lossy compression.

**Weaknesses:**

While the paper presents a novel and potentially impactful contribution, its clarity and accessibility are hindered by a dense presentation style. The heavy use of technical notation and the lack of illustrative examples make it challenging to grasp the core concepts and implications of the proposed framework.

Specifically, the paper would benefit from:

* More explanatory discussions: For instance, a concise discussion following Definition 3.3 would clarify the meaning and significance of the new formulation in comparison to the original RDP framework.

* Illustrative examples: Simple case studies or visual examples would help readers understand the practical implications of the theoretical results. The authors could consider drawing inspiration from the original RDP paper by Blau & Michaeli, which effectively uses examples to convey its ideas.

Addressing these issues would make the paper more accessible to a wider audience and increase its impact. While the core contribution merits acceptance, I strongly encourage the authors to revise the paper with a focus on clarity and illustrative examples.

**Questions:**

1. Does common randomness offer any benefits beyond perceptual realism in lossy compression? For example, stability/robustness?
2. How does the achievable rate-distortion-perception tradeoff change as a function of the batch size used by the universal critic?  Does this analysis offer any insights into selecting an appropriate batch size for training generative compression models that aim to satisfy perceptual quality constraints?

---

> ### Author Response · Authors · 2024-11-25
>
> Response to the “Weaknesses” section of the review:
>
> Following the reviewer’s advice, we provide more explanatory discussions regarding our
> proposed formulation, in comparison with the original RDP framework. The main
> difference with the latter pertains to the realism constraint. In the original formulation of
> Blau & Michaeli, the realism constraint is $\mathcal{D}(p_X, p_Y) \leq \lambda,$ where
> $\mathcal{D}$ is some divergence, $p_X$ is the source distribution, and $p_Y$ is the
> reconstruction distribution. In Definition 3.3, which introduces our proposed
> framework, we formulate a realism constraint involving a batch of $B$ reconstructions $
> Y^{(1)}, …, Y^{(B)},$ and a (lower semi-)computable critic $\delta$ inspecting said
> batch: $E[\delta(Y^{(1)}, …, Y^{(B)})] \leq C.$ Intuitively, the original realism constraint
> corresponds to the case where the batch size is infinite, since the discrete distributions
> $p_X$ and $p_Y$ can be approximated arbitrarily well using a large enough number of
> samples. In that sense, our proposed RDP framework generalizes the original one,
> through involving elements of practical realism metrics, such as the number $B$ of
> samples which are inspected, and a scoring function $\delta$ which is required to be
> approximable via an algorithm. Our results (Theorems 4.1 and 4.4) show that the choice
> of $B$ greatly impacts the RDP trade-off, especially the role of randomness, thereby
> questioning the original RDP framework.
>
> The reviewer also advised to provide illustrative examples. After careful consideration,
> we have reached the conclusion that the core concept of our paper, namely the
> optimality of deterministic codes, cannot be illustrated by examples. Indeed, we are not
> aware of any experimental method allowing to ascertain the optimality of a given
> compression algorithm (at a given target rate) for non-trivial choices of source, critic,
> and distortion measure. Only a theoretical proof can show the optimality of
> deterministic schemes. Moreover, to conclusively show that randomness is not
> necessary to fool critics, one must consider the most powerful critic possible, which
> cannot be practically implemented.
>
> Response to the “Questions” section of the review:
>
> 1. Our results show the existence of optimal schemes which do not involve any
> randomness at test time, but there may exist other optimal schemes, which rely
> on randomness at test time, as well as learned schemes relying on randomness
> at training time. We are not aware of any evidence that randomization in
> compression or decompression is necessary for optimal stability or robustness,
> but it is possible that such evidence will be found in the future.
>
> 2. Quantifying the dependence of the RDP trade-off on the batch size used by the
> critic is a difficult problem. While we have obtained results on the role of
> randomization, we do not yet have results which could guide the choice of the
> batch size for designing generative compression schemes.

---

> > ### Comment · Reviewer_omKD · 2024-12-02
> > **Reponse to Authors**
> >
> > I thank the authors for their response. I would like to clarify that illustrative examples may include toy examples and simple theoretical cases for which the optimal solution is known or can be derived.  I believe the addition of illustrative examples would strengthen the paper, and in their absence, I must maintain my current evaluation.

---

> > > ### Author Response · Authors · 2024-12-04
> > >
> > > Regarding simple theoretical examples, one of the simplest sources compatible with our finite-alphabet framework is the Bernoulli source. However, we have found the following difficulties. In our paper, we consider two asymptotic settings: a low batch size regime and a large batch size regime, which we shall call Setting 1 and Setting 2, for the sake of clarity. First, deriving an explicit solution for a Bernoulli source in a non-asymptotic setting seems to require new ideas. Second, this also seems to be the case for the optimal trade-off in Setting 2, except in the case of large common randomness rate. Third, the optimal trade-off in Setting 2, assuming large common randomness rate, happens to be identical to the trade-off in Setting 1 (where no common randomness is available). Therefore, the only illustration we could provide would be no more than a single rate-distortion curve (or two identical curves), which is a basic curve which has already appeared in the literature. Hence, we have not found any meaningful simple theoretical example.
> > >
> > > Given that the application area is of significant interest to the ML community, we believe our theoretical results would be of value to researchers in this community. It provides a fresh perspective to many results in the recent literature that deal with realism from an information theoretic perspective, and answers (partially) the question regarding the role common randomness plays in these results.
> > >
> > > Moreover, we would like to bring to the attention of the reviewer that many high-quality theoretical papers without any experiments or toy theoretical examples have appeared in machine learning conferences. The following are a few examples, among many:
> > >
> > > Unified Lower Bounds for Interactive High-dimensional Estimation under Information Constraints, NeurIPS 2023
> > >
> > > Distributed Learning with Sublinear Communication, ICML 2021
> > >
> > > Learning Policies with Zero or Bounded Constraint Violation for Constrained MDPs, NeurIPS 2021
> > >
> > > Optimal Algorithms for Non-Smooth Distributed Optimization in Networks, NeurIPS 2018

---

### Author Response · Authors · 2024-11-28
**Revised version of the paper**

Dear reviewers,

We have submitted a revised version of the paper. Following your suggestions, we have made three additions:

1. We have added a background section (Section 2.2) to further insist on the empirical evidence for our claims, namely state-of-the-art practical lossy compression schemes which do not make use of common randomness.
2. Directly following Definition 3.3, which introduces our new RDP trade-off, we have added a comparison to the original formulation by Blau and Michaeli, which our formulation generalizes, and highlighted the algorithmic nature of our realism constraint.
3. We have added a "Discussion" section at the end of the paper (Section 5), which includes the main take-aways regarding the role of the number of samples inspected by the critic, and avenues for future work.

We thank you for your valuable suggestions, which have allowed us to improve our paper.

---

### Meta-Review · Area_Chair_Q6E4 · 2024-12-20

**Metareview:**

Reviewers find the theoretical work interesting, but they agree that the paper lacks significant experimental demonstration of their results and also missing motivation for the utility of the presented work.
For that, the paper is recommended for rejection at this current state, but authors are encouraged to perform the experiments and resubmit.

**Additional Comments On Reviewer Discussion:**

Reviewers had a consensus about the lack of utility and validation. Authors response did not convince them.

---

### Decision · Program_Chairs · 2025-01-22

Reject